# OPTIMAS: OPTIMIZING COMPOUND AI SYSTEMS WITH GLOBALLY ALIGNED LOCAL REWARDS

**Shirley Wu**[*1], **Parth Sarthi**[*1], **Shiyu Zhao**[*1], **Aaron Lee**[1], **Herumb Shandilya**[1],
**Adrian Mladenic Grobelnik**[3], **Nurendra Choudhary**[2], **Eddie Huang**[2], **Karthik Subbian**[2],
**Linjun Zhang**[4], **Diyi Yang**[1], **James Zou**[**1], **Jure Leskovec**[**1]
[‡]Stanford University   [2]Amazon   [3]Jožef Stefan Institute   [4]Rutgers University

https://optimas.stanford.edu/

## ABSTRACT

Compound AI systems integrating multiple components, such as Large Language Models, specialized tools, and traditional machine learning models, are increasingly deployed to solve complex real-world tasks. However, optimizing compound systems remains challenging due to their non-differentiable structures and diverse configuration types across components, including prompts, hyperparameters, and model parameters. To address this challenge, we propose OPTIMAS, a unified framework for effective optimization of compound systems. The core idea of OPTIMAS is to maintain one *Local Reward Function* (LRF) per component, each satisfying a *local–global alignment* property, *i.e.,* each component's local reward correlates with the global system performance. In each iteration, OPTIMAS efficiently adapts the LRFs to maintain this property while simultaneously maximizing each component's local reward. This approach enables independent updates of heterogeneous configurations using the designated optimization method, while ensuring that local improvements consistently lead to performance gains. We present extensive evaluations across five real-world compound systems to demonstrate that OPTIMAS outperforms strong baselines by an average improvement of 11.92%, offering a general and effective approach for improving compound systems.

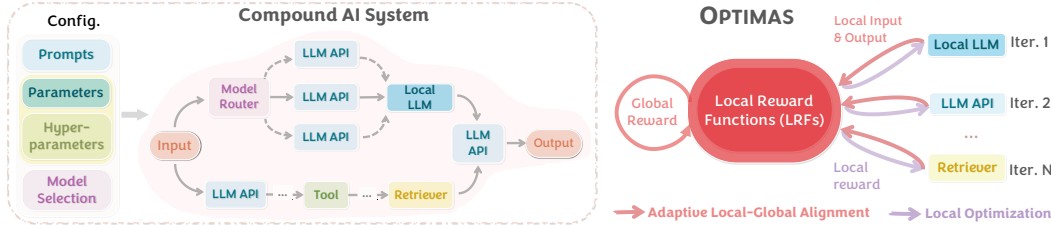

Figure 1: Overview. Given a compound AI system's heterogeneous configurations (*e.g.,* prompts, parameters) across multiple components, OPTIMAS maintains globally aligned *Local Reward Functions* (LRFs) as the system evolves, where each supervises a component and assigns higher local rewards to outputs with higher system performance (*aka.* global rewards). It iteratively adapts LRFs and optimizes each component to maximize its local reward for effective system optimization.

## 1 INTRODUCTION

Modern AI systems increasingly employ compound systems that integrate multiple complex components, such as Large Language Models (LLMs), tool/function calls, and traditional machine learning models like retrievers (Yuksekgonul et al., 2025; Khattab et al., 2023; Du et al., 2023). These components collaborate to process heterogeneous data sources and solve complex tasks through specialized subtask allocation (Zaharia et al., 2024; Zhou et al., 2025; Leike et al., 2018; Kandogan et al., 2025; Chen et al., 2025b). While compound AI systems have yielded performance advantages

---

[*]Equal contribution; [**]Equal supervision. Correspondence: <{shirwu,psarthi}@cs.stanford.edu>.

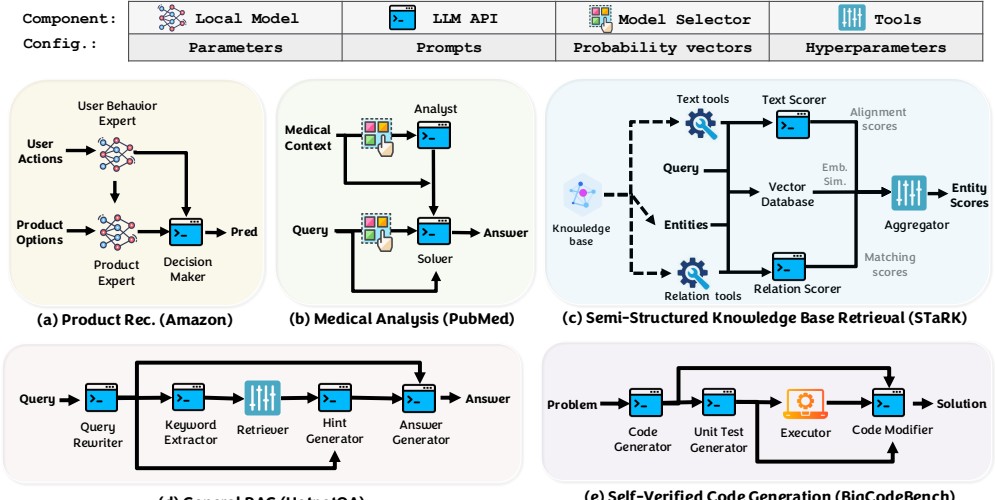

Figure 2: Five real-world and challenging compound AI systems. The goal is to automatically optimize the configuration across a heterogeneous set of components and parameters, *e.g.,* model parameters, prompts, model selection choice, and hyperparameters. See Appendix D for details.

over monolithic models (Du et al., 2023; Huang et al., 2023; Leblond & et al., 2023; Lewis et al., 2020; Liu et al., 2023), they can be highly sensitive to the failure of individual components, which leads to cascading failures in the final results (Cemri et al., 2025; tse Huang et al., 2025; Peng et al., 2025; Chen et al., 2024a). For example, if an LLM misinterprets an input query, it can retrieve irrelevant or misleading information. This leads to subsequent tool calls operating on incorrect inputs, producing unreliable outputs throughout the system. Therefore, optimizing these systems as a whole is crucial for maintaining reliability and global system performance (*i.e.,* global rewards).

However, optimizing these compound AI systems end-to-end is fundamentally challenging due to their non-differentiable nature. Second, it is hard to jointly optimize heterogeneous configurations (textual or numerical, continuous or discrete) from individual components, including prompts, hyperparameters, model selections, and even model weights. Moreover, running the entire compound AI system during optimization to achieve global reward is costly.

Previous works have largely focused on optimizing specific configurations in isolation, such as optimizing prompts through textual feedback (Yuksekgonul et al., 2025; Khattab et al., 2023; Yang et al., 2024; Madaan et al., 2023; Wu et al., 2024a) or model selection through iterative search (Chen et al., 2025b; 2024c). Yet these approaches can fail to capture critical bottlenecks. For example, a perfectly optimized prompt can struggle to compensate for a poorly chosen model. Even when the components are individually well optimized, they may still collaborate suboptimally, as the upstream component might not have visibility into which inputs are effective for the downstream components. Consequently, previous methods typically require costly system runs over many configurations to identify the best configuration for the components to work well together, leading to low data efficiency due to the large number of configurations.

**Present work**. Here we propose OPTIMAS (Figure 1), a unified framework for the effective and data-efficient optimization of compound AI systems. The core idea is to *learn* a **globally aligned** *Local Reward Function* (LRF) per component, such that independently maximizing a component's local reward still reliably improves the global rewards. We show that under mild conditions, our approach converges reliably, providing strong theoretical guarantees. Furthermore, since the learned LRFs can be used to optimize components locally, OPTIMAS has higher data efficiency by avoiding extensive runs of the entire compound AI system to achieve high global reward.

Specifically, each LRF estimates the contribution of a component's output to the global reward. All LRFs are implemented using a shared LLM backbone, with separate projection heads for each component to produce component-specific rewards. We propose a **lightweight adaptation** mechanism using mini-batch preference data to ensure the LRFs remain aligned with the evolving system configuration (Figure 3). Leveraging this decentralized structure, OPTIMAS applies specific

Table 1: A comparison of OPTIMAS with selected methods. OPTIMAS optimizes compound systems with heterogeneous configurations and enables higher data efficiency with local optimization to reduce number of system runs. We prove OPTIMAS's convergence under mild conditions.

| | Supports compound system | Optimizes heterogeneous config. | Data efficiency | Convergence guarantee |
|---|:---:|:---:|:---:|:---:|
| OPRO (Yang et al., 2024) | ✗ | ✗ | ✗ | ✗ |
| DSPy (Khattab et al., 2023) | ✔ | ✗ | ✗ | ✗ |
| TextGrad (Yuksekgonul et al., 2025) | ✔ | ✗ | ✗ | ✗ |
| LLMSelector (Chen et al., 2025b) | ✔ | ✗ | ✗ | ✔ |
| OPTIMAS | ✔ | ✔ | ✔ | ✔ |

optimization method to each component based on its configuration type. For example, reinforcement learning for model parameters (Rafailov et al., 2023; Schulman et al., 2017) or metric-guided search for prompts and hyperparameters (Yang et al., 2024; Opsahl-Ong et al., 2024; Liashchynskyi & Liashchynskyi, 2019). Overall, OPTIMAS iteratively updates heterogeneous configurations towards a higher global reward by using each adaptive LRF as an objective. By optimizing a component to maximize its local reward, OPTIMAS reduces the entire system runs to maintain higher data efficiency.

We conduct extensive experiments to evaluate OPTIMAS across five real-world compound systems (Figure 2), including challenging settings such as behavior-driven product recommendation and medical analysis. OPTIMAS consistently outperforms strong baselines, achieving an average relative improvement of 11.92% with higher data efficiency, while baseline methods occasionally improve performance. For example, while DSPy improves the performance on the multi-hop QA system, it may degrade performance on other tasks, such as product recommendation. In contrast, OPTIMAS is the *only* method that improves performance across *all* five tasks, consistent with our theoretical guarantee (Section 4.4) that aligning local and global rewards enables effective optimization.

## 2 RELATED WORK

**Optimizing LLM single-step generation**. Prior work extensively optimizes prompts for Large Language Models (LLMs) in single-step generation to improve performance (Yang et al., 2024; Madaan et al., 2023; Guo et al., 2024; Shinn et al., 2023; Yu et al., 2024; Yin et al., 2025; Chen et al., 2025a; Williams, 1992), but these methods are limited in their ability to handle complex, multi-step tasks. For example, addressing complex queries often requires combining multiple components—such as LLMs, tools, and machine learning predictors—to obtain accurate predictions.

**Optimizing multi-component/multi-step generation**. Compound AI systems consisting of multiple components enable more complex planning and specialized processing at each task step (Khattab et al., 2023; Yao et al., 2023; Liu et al., 2023; Du et al., 2023; Zhang et al., 2025). Previous studies typically optimize different components separately, such as optimizing LLM prompts (Khattab et al., 2023; Wu et al., 2024a; Yuksekgonul et al., 2025), fine-tuning model weights using supervised learning (Zhao et al., 2025; Chen et al., 2024d) or reinforcement learning (Lin et al., 2023; Chen et al., 2025c), developing model routing (Chen et al., 2025b) and layer grouping (Chen et al., 2023) strategies, and selecting hyperparameters (Wang et al., 2023; Falkner et al., 2018; Pham et al., 2018; Liu et al., 2019). In contrast, OPTIMAS enables end-to-end optimization across *all* components.

**Reward modeling for multi-step tasks**. To provide more fine-grained supervision, recent works break down global rewards (*e.g.,* answer accuracy) into more targeted signals (*i.e.,* dense/process rewards). Representative approaches include leveraging or bootstrapping from human step-wise annotations (Lightman et al., 2024; Uesato et al., 2022; She et al., 2025); hierarchical planning that assigns rewards to error correction steps (Wang et al., 2025); using Monte Carlo Tree Search to assign credit to intermediate reasoning steps (Wang et al., 2024; Ma et al., 2025; Jiao et al., 2024; Chen et al., 2024b; Setlur et al., 2025) or actions (Chen et al., 2025d; Choudhury, 2025). Recently, Chen et al. (2024d) leverage Bayesian optimization to decompose global losses into local losses for optimizing model weight. Optimas differs by dynamically aligning local rewards with global rewards through preference-based adaptation. This design is applicable to both differentiable and non-differentiable configurations, without requiring fixed decomposition or extensive retraining. The local optimization avoids extensive system runs and offers higher data efficiency. Moreover, we provide theoretical analysis to prove the convergence of our framework. We highlight our key contributions in Table 1.

## 3   PROBLEM FORMULATION: OPTIMIZING COMPOUND AI SYSTEMS

**Compound AI system**. A compound AI system is represented as a directed acyclic graph $\mathcal{G} = (\mathcal{C}, \mathcal{E})$, where $\mathcal{C} = \{C_k\}_{k=1}^K$ is a set of $K$ distinct components (task nodes) and $\mathcal{E}$ is the set of all possible directed edges between components. A component of the compound system can be an LLM, a general machine learning model, a model selector, *etc*. We denote the input and output to each component $C_k$ as $x_k$ and $y_k$, respectively. The system input is treated as a source node $C_0$.

The system can operate with dynamic planning: for each input instance $x$, the connections $\mathcal{E}(x) \subseteq \mathcal{E}$ between the components can be adaptive. A directed edge $(C_i, C_j) \in \mathcal{E}(x)$ indicates that the output of component $C_i$ is routed as input to component $C_j$ when processing instance $x$. By default, we assume the component indices follow the topological order over $\mathcal{E}$, meaning that $C_i$ is the upstream component of $C_j$ if $i < j$.

**Component configurations**. A component $C_k : (x_k; \mathbf{v}_k) \mapsto y_k$ is controlled by a configuration policy $\mathbf{v}_k$. The configuration space $\mathcal{V}$ can either be empty (indicating no optimizable configuration for the component), discrete (*e.g.,* textual prompts or model selections), or continuous (*e.g.,* model parameters or hyperparameters). We denote the joint configuration policy by $\mathbf{v} = (\mathbf{v}_1, \ldots, \mathbf{v}_K)$.

**Forward execution**. For a given input $x$ and configuration policy $\mathbf{v}$, the system executes components in topological order over the edge set $\mathcal{E}(x)$: $y_k = C_k(\{y_i \mid C_i \in pa(C_k)\}; \mathbf{v}_k)$, where $pa(C_k)$ denotes all the parents of component $C_k$ over $\mathcal{E}(x)$. For clarity, we define the overall system as $f(x; \mathbf{v}) := y$, where $y$ is a collection of outputs from one or more components.

**Optimization objective**. Given a dataset $\mathcal{D}$ with initial inputs and a user-defined global reward function $R : \mathcal{X} \times \mathcal{Y} \to \mathbb{R}$ that evaluates the final system output, the optimization goal is to find the configuration policy $\mathbf{v}^\star(x)$ that maximizes the expected global reward:

$$\mathbf{v}^\star = \arg\max_{\mathbf{v}} \ \mathbb{E}_{x \sim \mathcal{D}}\big[R(x, f(x; \mathbf{v}))\big]. \tag{1}$$

## 4   OPTIMAS: GLOBALLY ALIGNED LOCAL REWARDS FOR OPTIMIZATION

**Challenges**. Directly optimizing the objective in Eq. 1 is difficult. As the configuration spaces are typically non-differentiable, gradient-based optimization cannot be used. Moreover, each policy $\mathbf{v}_k$ may control a different configuration type, so the joint policy $\mathbf{v}$ can span heterogeneous spaces. Therefore, previous efforts (Yuksekgonul et al., 2025; Zhao et al., 2025; Chen et al., 2025b; Khattab et al., 2023) largely focus on optimizing the policy for single types of configurations, which simplifies the optimization problem; however, this also leads to suboptimal compound systems.

**Key intuition**. To address the challenges, our approach (Figure 3) learns Local Reward Functions (LRFs) that align with the global reward for individual components, allowing local and independent optimization on heterogeneous components using different optimization approaches.

Such **local-global alignments** (Section 4.1) encourage that the global reward to increase during local optimizations (Section 4.3). Moreover, as the system configurations change during optimization, the LRFs should be adapted to remain aligned. To ensure alignment, OPTIMAS employs a **lightweight adaptation** mechanism that updates LRFs with minimal data sampled from the system, preserving consistency with the global reward (Section 4.2).

### 4.1   LEARNING LOCAL REWARD FUNCTIONS

**Definition (Local Reward Function (LRF))**. An LRF on component $C_k$ is defined as $r_k : (x_k, y_k) \to \mathbb{R}$, which evaluates the component's output $y_k$ given the provided context $x_k$.

**Implementation**. We implement all LRFs with a LLM backbone $\phi$ and separate linear heads $h_k$ for a component $C_k$. The backbone encodes the concatenated text inputs $[x_k, y_k]$ into an embedding, and the corresponding head projects this embedding to a scalar reward value. Using such a multitask neural network ensures scalability with large number of components and reduces memory costs. Specifically, each LRF is modeled as:

$$r_k(x_k, y_k) = h_k \circ \phi([x_k, y_k])), \quad \text{for all } k \text{ if } \mathbf{v}_k \text{ is non-empty.} \tag{2}$$

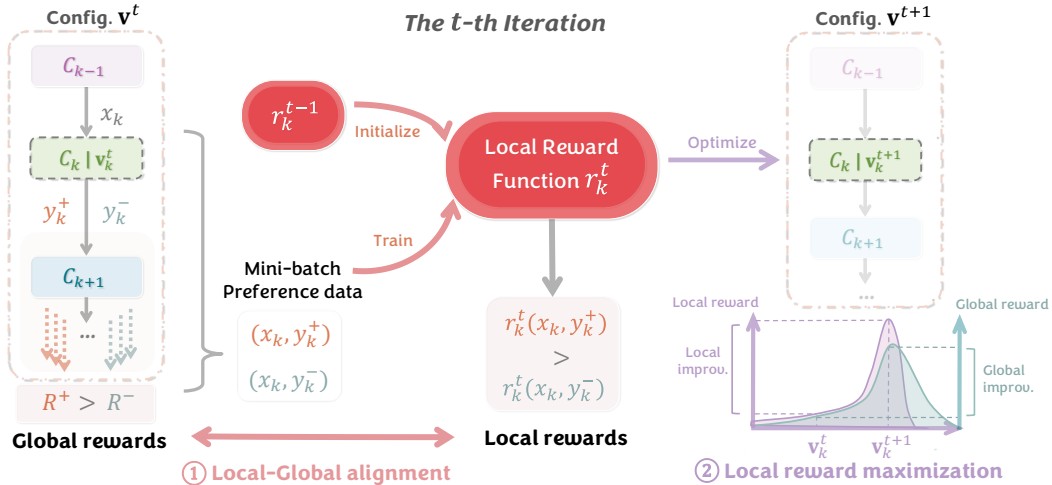

Figure 3: OPTIMAS optimization iteration. At each iteration, OPTIMAS updates a component $C_k$ by first collecting a mini-batch of preference data and adapting its Local Reward Function $r_k$ to remain aligned with the global task metric. This alignment helps ensure that optimizing the component to maximize its local reward also improves the global reward.

**Property (Local–global alignment).** An LRF $r_k$ is said to be *aligned* with the global reward $R$ if, for every input $x$ and for any two candidate outputs $y_k^+, y_k^-$ of $C_k$,

$$
\begin{aligned}
& r_k(x_k, y_k^+) \geq r_k(x_k, y_k^-) \\
& \implies \mathbb{E}_{\text{downstream}}\Big[R\big(x, f(x; \mathbf{v}_{-k})\big) \mid y_k^+\Big] \geq \mathbb{E}_{\text{downstream}}\Big[R\big(x, f(x; \mathbf{v}_{-k})\big) \mid y_k^-\Big],
\end{aligned}
\tag{3}
$$

where $\mathbf{v}_{-k}$ denotes the configurations of all *downstream* components (those that directly or indirectly receive information originating from $C_k$). The expected global reward for each candidate output is estimated via Monte Carlo sampling. This involves executing the downstream components with the candidate output and the outputs from the non-downstream components fixed, capturing their stochasticity, and averaging the resulting global rewards from the final system outputs.

**Objective of reward functions.** To make each LRF $r_k$ conform to the local-global alignment property, we collect $\mathcal{D}_k(\mathbf{v})$, a preference dataset under the current system configuration $\mathbf{v}$, and train each $r_k$ using a pairwise log-sigmoid ranking loss:

$$
\mathcal{L}_k(\mathcal{D}_k(\mathbf{v})) = -\mathbb{E}_{(x_k, y_k^+, y_k^-) \sim \mathcal{D}_k(\mathbf{v})}\Big[\log \sigma\big(r_k(x_k, y_k^+) - r_k(x_k, y_k^-)\big)\Big],
\tag{4}
$$

The collection of $\mathcal{D}_k(\mathbf{v})$ follows the following steps: 1) execute the compound system up to $C_k$ and record the partial trajectory $\langle x, (x_1, y_1), \ldots, (x_{k-1}, y_{k-1}) \rangle$; 2) sample two candidate outputs for $C_k$ (*e.g.,* via higher-temperature decoding or alternate hyperparameters); and 3) estimate their expected task metrics according to the expectation terms on the right-hand side of Eq. 3. The output with the higher expected value is labeled as $y_k^+$, and the other as $y_k^-$ in $\mathcal{D}_k(\mathbf{v})$.

## 4.2 ADAPTIVE LOCAL REWARD FUNCTIONS

**Problem: Misaligned LRFs in the evolving system.** As the system configuration changes during optimization, LRFs trained under a previous configuration $\mathbf{v}^t$ may become inaccurate under the updated configuration $\mathbf{v}^{t+1}$. Specifically, (1) after updating $C_k$, the same outputs from its upstream component $C_i$ $(i < k)$ may lead to different global reward, making $r_i$ misaligned. (2) Its downstream components $C_j$ $(j > k)$ now receive inputs generated by the updated $C_k$, which may fall outside the distribution seen by their LRFs. These shifts accumulate over time, degrading the local–global alignment property (Eq. 3) that LRFs are designed to satisfy.

However, retraining all LRFs from scratch after every configuration update is expensive. To address this, we develop a lightweight adaptation strategy that incrementally refines the LRFs as the system changes, maintaining alignment without full retraining.

**Stage 1: Initial reward modeling**. Given the initial system configuration and a dataset with initial inputs, we first construct an offline preference dataset for each component and train its LRF to convergence. This offline phase establishes well-aligned LRFs that accurately reflect each component's contribution to the global reward.

**Stage 2: Online reward function adaptation**. When any configuration changes, we sample a small batch of input data and construct a mini-batch of preference data $\mathcal{B}_k$ for each component $C_k$ using the steps described in Section 4.1. We then optimize the LRF on $C_k$ on the objective $\mathcal{L}_k(\mathcal{B}_k)$ following the definition in Eq. 4. To enable stable optimization and improve data efficiency, we maintain a buffer of previous generated preference data into $\mathcal{B}_k$. This adaptation helps maintain the local–global alignment property in Eq. 3.

## 4.3 OPTIMIZATION WITH GLOBALLY ALIGNED LOCAL REWARD FUNCTIONS

**Local Optimization**. As each component has its own LRF, OPTIMAS flexibly applies a specialized optimization method for each component. See details in Appendix F. Concisely,

- For textual prompts, we use prompt optimization algorithms such as OPRO (Yang et al., 2024) that ranks candidate prompts by their average local reward and select the best-performing one.

- For components that are trainable models (*e.g.,* an LLM), we apply reinforcement learning—such as Proximal Policy Optimization (PPO) (Schulman et al., 2017)—using the LRF as the critic.

- For discrete or low-dimensional continuous configurations, such as model selection or hyperparameter tuning, we construct a probability distribution over candidate values based on their local rewards and sample from it to update the configuration.

**Overall algorithm (Figure 3)**. Starting with the initial system configuration, OPTIMAS leverages initial reward modeling to learn a set of LRFs that are well-aligned with the global reward. At each iteration optimization, OPTIMAS randomly selects a component to optimize, conducts local optimization, and if the local configuration change leads to improved global reward, it updates the system and adapts the LRF using minimal amount of data. To prevent potential cascading errors, the new configuration is accepted only if it improves the global reward on a small validation set. Since the optimization with LRFs is conducted locally, the number of system runs to achieve a high global reward is reduced, as we later show in the experiments. The detailed algorithm is provided in Appendix A.

## 4.4 THEORETICAL INSIGHTS

We prove that the local–global alignment property holds for the LRFs constructed in Section 4.1.

**Theorem 4.1** (Informal)**.** *Under regularity conditions, the minimizer of equation 4 satisfies the local-global alignment property (equation 3). In addition, maximizing $r_k\big(x_k, C_k(x_k; \mathbf{v}_k)\big)$ over $\mathbf{v}_k$ and maximizing $R(x, f(x; \mathbf{v}_{-k}) \mid C_k(x_k; \mathbf{v}_k))$ over $\mathbf{v}_k$ will yield the same solution.*

We defer the formal statement for this theorem to appendix B. Since solving equation 1 is generally challenging, we introduce some regularity conditions to make the convergence analysis tractable.

As the configurations $\mathbf{v}$ are heterogeneous, where some of the coordinates are discrete, and some are continuous, without loss of generality, we assume the first $M$ configurations $\mathbf{v}^{(1)} = \{\mathbf{v}_1, ..., \mathbf{v}_M\}$ are continuous, and the last $(K - M)$ configurations $\mathbf{v}^{(2)} = \{\mathbf{v}_{M+1}, ..., \mathbf{v}_K\}$ are discrete. We write the objective function $\mathbb{E}_{x \sim \mathcal{D}}\big[R(x, f(x; \mathbf{v}))\big]$ as $l(\mathbf{v}) = l(\mathbf{v}_1, \mathbf{v}_2, ..., \mathbf{v}_K) := l(\mathbf{v}^{(1)}, \mathbf{v}^{(2)})$.

**Assumption 4.1.** *Suppose for any given configuration $\mathbf{v}^{(2)}$, the initial level set $\{\mathbf{v}^{(2)} : l(\mathbf{v}^{(1)}, \mathbf{v}^{(2)}) \leq l(\mathbf{v}^{0,(1)}, \mathbf{v}^{(2)})\}$ is a compact set, where $\mathbf{v}^{0,(1)}$ is the initialization used in the algorithm for $\mathbf{v}^{(1)}$. In addition, for every component $k$ and every fixed $\mathbf{v}_{-k}$, $l(\cdot, \mathbf{v}_{-k})$ has a unique maximizer.*

**Theorem 4.2.** *Under Assumption 4.1, the algorithm will converge to the component-wise maximum, that is, the limit point $\mathbf{v}^*$ satisfies*

$$l(\mathbf{v}^*) \geq l(\mathbf{v}_k, \mathbf{v}_{-k}^*),$$

*for any $k \in [K]$ and any $\mathbf{v}_k$.*

Table 2: Performances of each method on the compound systems. The best and second-best results in each column are highlighted. Relative improvement is computed with respect to the best baseline.

|  | AMAZON Product Rec. (Acc.) | PUBMEDQA Medical Analysis (Acc.) | STARK-PRIME Complex Retrieval (MRR) | HOTPOTQA RAG (F1) | BIGCODEBENCH Verified Code Gen. (Pass Rate) |
|---|---|---|---|---|---|
| Single LLM | 20.20±1.43 | 54.13±2.73 | 0.00±0.00 | 21.58±1.24 | 35.47±0.34 |
| Unoptimized | 21.21±3.78 | 57.46±0.75 | 40.73±0.64 | 33.80±1.51 | 36.67±1.35 |
| REINFORCE | 21.89±2.65 | - | - | - | - |
| LLMSelector | - | 67.93±0.09 | - | - | - |
| HBC | 21.55±2.07 | 58.80±0.58 | 36.95±0.59 | 21.16±0.97 | 27.78±2.08 |
| TextGrad | 20.88±3.53 | 56.96±2.24 | 41.31±1.67 | 24.86±1.19 | 35.71±0.10 |
| DSPy | 18.18±0.82 | 60.26±0.40 | 41.40±0.04 | 44.90±0.32 | 33.81±2.75 |
| OPTIMAS | **24.24±0.82** | **69.13±0.33** | **50.54±0.70** | **50.48±1.48** | **38.92±0.36** |
| Rel. Improv. | 14.3% | 1.8% | 22.1% | 12.4% | 9.0% |

Table 3: Number of equivalent runs on the entire systems (in thousands) by TextGrad, DSPy, and OPTIMAS in Table 2. We control the optimization process of each to use comparable system runs.

|  | AMAZON | PUBMEDQA | STARK-PRIME | HOTPOTQA | BIGCODEBENCH | Average |
|---|---|---|---|---|---|---|
| TextGrad | 0.32 | 0.70 | 0.70 | 2.12 | 0.18 | 0.80 |
| DSPy | 0.24 | 0.66 | 0.66 | 2.09 | 0.28 | 0.79 |
| OPTIMAS | 0.31 | 0.52 | 0.51 | 2.02 | 0.21 | 0.71 |

In fact, our theoretical analysis shows that by conducting local optimization, OPTIMAS is essentially performing coordinate maximization. Therefore, existing convergence results for coordinate maximization directly apply. Also, note that the block-coordinate (round-robin) updates adopted in OPTIMAS do not guarantee global optimality in non-convex problems, but this does not constitute a flaw unique to our method; rather, it reflects a standard limitation of non-convex optimization broadly, and our global convergence guarantees only hold under additional structural assumptions, such as Polyak–Łojasiewicz or Kurdyka–Łojasiewicz conditions.

## 5 EXPERIMENTS

We firstly summarize the datasets, baselines, and metrics, with details provided in the appendix.

**Benchmarks & Evaluation** (Appendix C). We evaluate OPTIMAS on five real-world tasks:

- **AMAZON** (Jin et al., 2024): A behavior-driven recommendation task based on Amazon products, evaluated using accuracy to measure if the predicted next item matches the ground-truth item.

- **PUBMEDQA** (Jin et al., 2019): A clinical classification dataset derived from PubMed abstracts (National Center for Biotechnology Information (NCBI), 2024), evaluated by accuracy, defined as the proportion of predictions that exactly match the ground-truth labels.

- **STARK-PRIME** (Wu et al., 2024b): A retrieval benchmark over semi-structured biomedical corpora, evaluated using Mean Reciprocal Rank (MRR).

- **HOTPOTQA** (Yang et al., 2018): A multi-hop question answering dataset, evaluated using the F1 score between predicted and ground-truth answers.

- **BIGCODEBENCH** (Zhuo et al., 2024): The instruction split of BigCodeBench for self-verifying code generation, evaluated using pass rate.

**Compound Systems** (Figure 2, *cf.* Appendix D). We design a compound system per benchmark with diverse and common patterns for agentic systems. All systems are accessible in our code repository.

**Baselines** (See Appendix E for details). We compare OPTIMAS against five baselines: Unoptimized, REINFORCE (Williams, 1992), LLMSelector (Chen et al., 2025b), Hierarchical Behavior Cloning (HBC) (Le et al., 2018), TextGrad (Yuksekgonul et al., 2025), DSPy (Khattab et al., 2023; Opsahl-Ong et al., 2024). Moreover, we provide a single LLM reference which prompts an LLM to complete the task directly. This reference is used for justifying our system design in the experimental setup.

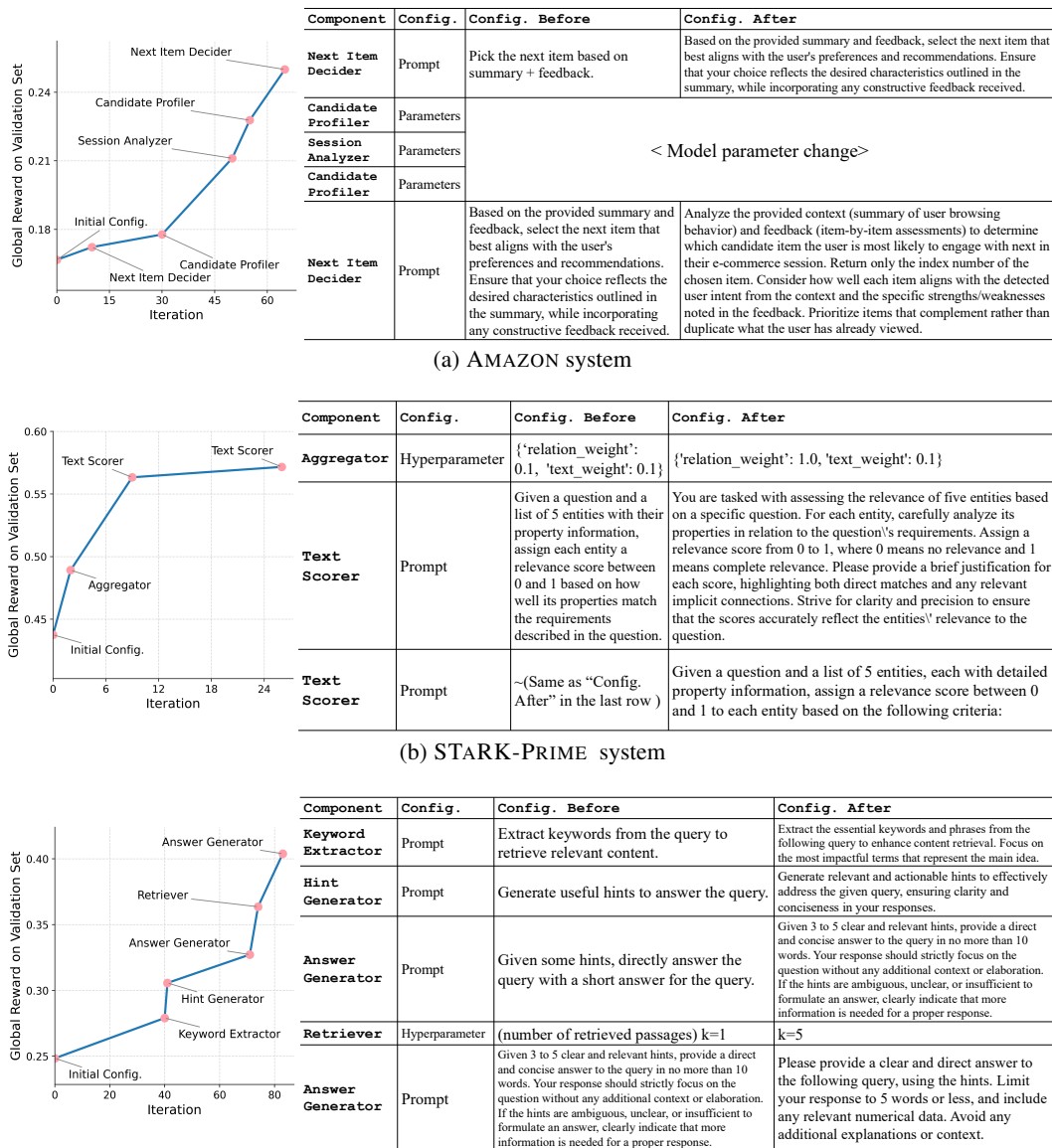

Figure 4: Global reward and configuration updates of the three compound AI systems over the optimization iterations. For conciseness, we only show the local optimization steps that lead to an increase in global reward on the validation sets. The annotations show the optimized components.

Table 4: Average pairwise ranking accuracy on validation sets, measuring how often the method assigns a higher score to the output with higher expected global reward.

|  | AMAZON | PUBMEDQA | STARK-PRIME | HOTPOTQA | BIGCODEBENCH | Avg. |
|---|---|---|---|---|---|---|
| LLM Judge | 51.25% | 49.54% | 54.37% | 50.00% | 42.45% | 49.52% |
| OPTIMAS | 84.93% | 65.28% | 76.64% | 72.40% | 90.57% | 77.96% |

## 5.1 PERFORMANCE OF THE SYSTEMS DURING AND AFTER OPTIMIZATION.

**Takeaway 1: OPTIMAS leads to consistent and substantial improvements (Table 2).** We compare OPTIMAS with the baselines under similar numbers of system runs. Under this controlled data cost (Table 3), OPTIMAS consistently improves global rewards across all compound systems, achieving an average relative improvement of 11.92% compared to the best baseline.

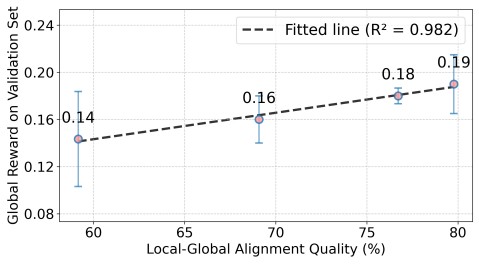
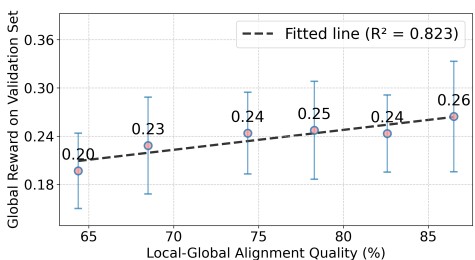

(a) `Candidate Profiler` on AMAZON  (b) `Answer Generator` on HOTPOTQA

Figure 5: Local reward models with varying alignment quality are used to optimize a selected component in each task, where we observe that higher alignment quality yields higher global rewards.

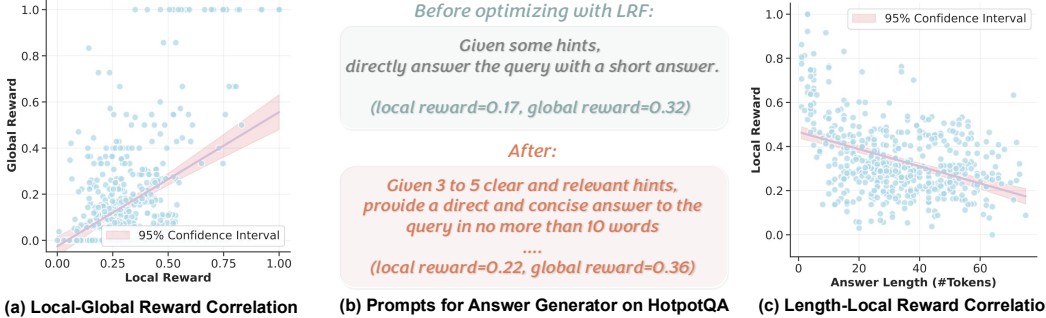

(a) Local-Global Reward Correlation  (b) Prompts for Answer Generator on HotpotQA  (c) Length-Local Reward Correlation

Figure 6: An interpretability study on what is learned by LRFs. With (a) a well-aligned LRF, we find that (b) the optimized prompt explicitly constrains the output length of the component. We attribute this to that (c) the LRF prefers short outputs, which is consistent with the use of F1 as global metric.

For REINFORCE, while it yields performance gain from the unoptimized baseline, OPTIMAS outperforms it by around 3%. Moreover, REINFORCE needs collecting the reward signal from downstream Monte Carlo sampling, which requires more than three times the data than OPTIMAS. While the strong baseline DSPy shows notable improvements on some datasets (*e.g.,* a 21.6% gain on HOTPOTQA), their performance may be inconsistent and can even degrade the system (*e.g.,* a 14.3% drop on the AMAZON dataset). For LLMSelector, it requires $2.8k$ times of forwarding through the entire system, which is $3x$ more expensive than OPTIMAS.

**Takeaway 2: Local optimization improves global rewards (Figure 4).** We study how the configurations change in the local optimization. Within a small number of iterations, OPTIMAS achieves a substantial average improvement of 41.7% over the initial global reward on the validation sets, using lightweight and data-efficient updates. Interestingly, we observe a mixed updates on prompts, model parameters, and hyperparameters, which can lead to improved global reward. For example, updating the prompt of `Text Scorer` in the 9-th iteration improves global reward from $0.49$ to $0.56$. Among these cases, they involve optimizing different components to achieve the highest global reward empirically, showing the importance of being able to optimize different types of configurations.

### 5.2 WHY AND HOW OPTIMAS WORKS: ALIGNMENT, INTERPRETABILITY, AND EFFICIENCY

We conduct extensive in-depth study to understand the mechanism in OPTIMAS framework.

**Takeaway 3: OPTIMAS yields high local-global alignment quality (Table 4).** To measure alignment quality of LRFs, we compute pairwise ranking accuracy: the probability that an output with higher global reward receives a higher score than an output with lower global reward. This reflects how well the learned LRFs aligns with global rewards. We compare against a LLM Judge, which prompts a `gpt-4o` model to score the outputs of components based on 20 in-context examples. This approach is similar to prior methods such as TextGrad, which rely on few-shot reasoning over textual patterns. In Table 4, LLM Judge performs closer to random guessing, due to the diversity and stochasticity of components' outputs that make it difficult to reason reliably. In contrast, our LRFs achieve substantially higher performances. Moreover, LRFs internalize the local-global alignment within their weights without relying on limited in-context examples, enabling more precise alignment.

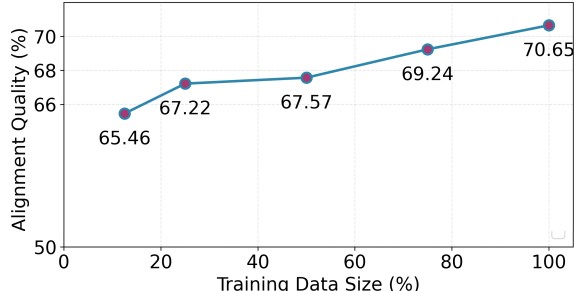

| Backbone size | Alignment quality (%) |
|---|---|
| 1B | 72.58 |
| 3B | 70.04 |
| 8B | 71.20 |

Figure 7: Impact of training data size and LRF backbone size on alignment quality, measured by pairwise ranking accuracy: the percentage of preference pairs where local and global rewards agree.

**Takeaway 4: Higher alignment quality usually leads to higher global reward (Figure 5).** To understand how alignment quality affects global reward, we conduct controlled experiments, where we select a key component, apply local reward models with varying alignment quality to optimize it, and measure the average global reward achieved after updating the component. In Figure 5, we show a strong positive correlation between the LRF's alignment quality and the global reward improvement.

**Takeaway 5: LRFs learn interpretable directions to improve global reward (Figure 6).** An important aspect of system optimization is interpretability, *i.e.,* if the configuration updates are reliable and understandable. We provide a study in Figure 6, where the LRF learns to favor concise answers. In fact, it is more feasible to interpret configuration updates with LRFs. Specifically, one can perturb the component outputs in certain ways, and observe the changes in local rewards to obtain insights.

**Takeaway 6: OPTIMAS is data- and computationally efficient. (Figure 7 & Appendix G).** We study the efficiency of OPTIMAS along two axes on the HOTPOTQA system: (1) how much training data is needed to learn effective LRFs, and (2) whether large LRF backbones are necessary.

We train LRF models using varying percentages of available training data (12.5%–100%) and measure alignment quality via pairwise ranking accuracy. Figure 7 (left) shows the performance degrades gracefully: using just 12.5% of the data yields 65.46% accuracy (92.7% of full-data performance), and 25% recovers 95.1%. This indicates that OPTIMAS is **data efficient**, where LRF can be learned from relatively few system runs. Moreover, Figure 7 (right) compares 1B, 3B, and 8B LRF backbones and reports alignment quality. The results show that OPTIMAS is **computationally efficient**, where lightweight models are sufficient for learning local-global alignment.

In Appendix G, we further show that local and global reward landscapes are closely aligned across retriever top-$k$ settings, that modest numbers of prompt candidates and adaptation inputs suffice for strong performance, and provide a cost comparison on the AMAZON system showing that OPTIMAS achieves the best performance while using a comparable number of effective system runs.

## 6   CONCLUSION

OPTIMAS is a unified framework to optimize compound AI systems with heterogeneous configurations. OPTIMAS' way to maintain globally aligned local reward functions allows every component, whether a fine-tunable LLM, LLM API, tools, or model selector, to be optimized locally while improving the overall system. On five real-world tasks, OPTIMAS outperforms strong baselines, effectively optimizes components with different configurations, and exhibits high alignment quality and reliable interpretations. We believe OPTIMAS will serve as a general, data-efficient approach for continually optimizing practical systems. For future work, we aim to further apply OPTIMAS on even larger systems, with the goal of understanding complex reward modeling and scalability.

## ACKNOWLEDGEMENTS

We gratefully acknowledge the support of Amazon, NSF under Nos. CCF-1918940 (Expeditions), DMS-2327709 (IHBEM), IIS-2403318 (III); NIH under No. 1U24NS146314-01, Stanford Data Applications Initiative, Wu Tsai Neurosciences Institute, Stanford Institute for Human-Centered AI, Chan Zuckerberg Initiative, Genentech, SAP, and SCBX.

## ETHICS STATEMENT

The applications of compound AI systems today include, but are not limited to, decision processes in socially and economically important domains, making their optimization an increasingly significant research problem. While our work aims to make optimization of such systems more data-efficient, robust, and interpretable, we recognize potential ethical concerns. In high-stakes settings such as healthcare or recommendation platforms, optimization could inadvertently amplify biases or propagate unsafe behaviors from individual components; our framework mitigates these risks by aligning local rewards with global performance, thereby reducing the likelihood of harmful emergent behaviors and creating opportunities to incorporate fairness- and safety-sensitive objectives. Another concern is malicious use of optimized compound systems; however, our approach relies on safety-aligned LLMs and tools, and we observe that aligning local and global objectives does not reduce the effectiveness of existing safeguards, while the decentralized optimization structure may even increase opportunities to detect misuse. Regarding data, our experiments rely solely on publicly available datasets or synthetic user interactions without any personally identifiable information (PII), and no private or sensitive user data were used. We believe this work contributes to the safe and responsible advancement of AI by providing both theoretical and practical insights into optimizing compound AI systems, and to promote further research in this direction we release all code, models, and benchmarks described in this paper.

## REPRODUCIBILITY STATEMENT

We provide full open access to our implementation code, including the original datasets, compound system implementations, and the OPTIMAS framework. To support reproducibility, we provide Appendix C (Dataset Details), Appendix D (System Details), Appendix E (Baseline Details), and finally, Appendix F (OPTIMAS Details).

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

# A  ALGORITHM

---

**Algorithm 1** Component-wise optimization with reward-model adaptation

---

**Input**
$\mathcal{C} = \{C_1, \ldots, C_K\}$ ▷ components
$\mathbf{v}^0 = (\mathbf{v}_1^0, \ldots, \mathbf{v}_K^0)$ ▷ initial configuration policy
$\Theta^0 = \{\theta_1^0, \ldots, \theta_K^0\}$ ▷ parameters of local reward functions
Global reward $R(\cdot)$, preference dataset size $k$, total iterations $T$
Training dataset $\mathcal{D}$ and validation dataset $D_v$
1: $\mathbf{v}^* \leftarrow \mathbf{v}^0$
2: $\Theta^t \leftarrow \Theta^0$
**Optimization loop**
3: **for** $t = 0, \ldots, T-1$ **do**
**Scheduler: choose a component to optimize**
4:      $i_t \sim \text{Uniform}(\{1, \ldots, K\})$
**Local optimization for the chosen component** $C_{i_t}$
5:      **if** $C_{i_t}$ is an LLM with prompts **then**
6:         $\tilde{\mathbf{v}}_{i_t}^{t+1} \leftarrow \text{PROMPTOPTIMIZATION}(\mathbf{v}_{i_t}^t, \theta_{i_t}^{t+1})$
7:      **else if** $C_{i_t}$ has trainable weights **then**
8:         $\tilde{\mathbf{v}}_{i_t}^{t+1} \leftarrow \text{PPOTRAIN}(\mathbf{v}_{i_t}^t, \theta_{i_t}^{t+1})$
9:      **else if** $C_{i_t}$ has a hyperparameter configuration **then**
10:     $\tilde{\mathbf{v}}_{i_t}^{t+1} \leftarrow \text{HYPERPARAMETERSEARCH}(\mathbf{v}_{i_t}^t, \theta_{i_t}^{t+1})$
11:    **end if**
12:    $\tilde{\mathbf{v}} \leftarrow \mathbf{v}^t$ with $\mathbf{v}_{i_t}^t$ replaced by $\tilde{\mathbf{v}}_{i_t}^{t+1}$
**Validation**
13:    **if** $\sum_{x_v \in D_v} R(x_v, f(x_v; \tilde{\mathbf{v}})) > \sum_{x_v \in D_v} R(x_v, f(x_v; \mathbf{v}^t))$ **then**
14:      $\mathbf{v}^{t+1} \leftarrow \tilde{\mathbf{v}}; \quad \mathbf{v}^* \leftarrow \mathbf{v}^{t+1}$
15:      **Reward-model adaptation**
16:      $D_t \leftarrow \text{COLLECTPREFERENCEDATA}(\mathcal{D}, \Theta^t, k)$ ▷ create $k$ $(x^+, x^-)$ pairs
17:      $\Theta^{t+1} \leftarrow \text{REWARDMODELTRAIN}(\Theta^t, D_t)$
18:    **else**
19:      $\mathbf{v}^{t+1} \leftarrow \mathbf{v}^t$
20:      $\Theta^{t+1} \leftarrow \Theta^t$
21:    **end if**
22: **end for**
23: **return** $\mathbf{v}^*$

---

# B  THEORETICAL ANALYSIS

**Formal statement of and proof of Theorem 4.1** According to the procedure described in Section 4, the positive and negative pairs are determined by comparing the expected task metrics. We assume the estimated metrics at $y_k^+$ and $y_k^-$ are chosen following $\mathbb{P}(y_k^+ \text{ is labeled as positive}) = \sigma_\alpha(\mathbb{E}_{\text{downstream}}\left[R(x, f(x; \mathbf{v}_{-k}(x))) \mid y_k^+\right] - \mathbb{E}_{\text{downstream}}\left[R(x, f(x; \mathbf{v}_{-k}(x))) \mid y_k^-\right])$, where $\sigma_\alpha(u) = \frac{1}{1 + \exp(-\alpha u)}$ is the sigmoid function with parameter $\alpha > 0$. $\alpha = +\infty$ corresponds to the case where the pairs are chosen deterministically.

**Theorem B.1.** *Under the conditions specified above, the maximizer of Eq. 4 satisfies the local-global alignment property Eq. 3. In addition, maximizing $r_k(x, C_k(x_k; \mathbf{v}_k))$ over $v_k$ and maximizing $R(x, f(x; \mathbf{v}_{-k}) \mid C_k(x_k; \mathbf{v}_k))$ over $v_k$ will yield the same solution.*

*Proof.* We first present a lemma.

**Lemma B.1.** *Suppose $(\boldsymbol{x}, y) \in \mathbb{R}^p \times \{-1, 1\}$ follows the distribution $\mathbb{P}(y = 1 \mid \boldsymbol{x}) = \sigma_1(p^*(x))$ for some function $p : \mathbb{R}^p \to (0, 1)$. Then*

$$\arg\max_p \mathbb{E}[\log(\sigma_1(y \cdot p(x)))] = p^*.$$

*Proof.* We take the derivative of the left-hand side with respect to $p$ and set it to 0:

$$\mathbb{E}\big[\frac{\sigma_1(y \cdot p(x)) \cdot \sigma_1(-y \cdot p(x))}{\sigma_1(y \cdot p(x))} \cdot y\big] = 0,$$

which is equivalent to

$$\mathbb{E}[\sigma_1(-y \cdot p(x)) \cdot y] = 0.$$

We then have

$$\sigma_1(-p(x))\sigma_1(p^*(x)) - \sigma_1(p(x))\sigma_1(-p^*(x)) = 0,$$

and therefore $p(x) = p^*(x)$. $\qquad\square$

Applying this lemma, we can obtain the solution to Eq. 4 is $\alpha \cdot \mathbb{E}_{\text{downstream}}\Big[R\big(x, f(x; \mathbf{v}_{-k}(x))\big) \,\big|\, y_k\Big]$ for some positive $\alpha$, and therefore it satisfies the local-global alignment property Eq. 3.

In the following, we then prove that maximizing $r_k(x, \pi_k(x))$ over $v_k$ and maximizing $R(x, f(x; \mathbf{v}_{-k}) \,|\, \pi_k(x))$ over $v_k$ will yield the same solution.

**Lemma B.2.** *Assume the local–global alignment property Eq. 3 holds for component $C_k$. Let $\mathbf{v}(x)$ and $\tilde{\mathbf{v}}(x)$ be two configuration policies that differ* only *in the policy for component $C_k$; denote the corresponding local outputs by $y_k$ and $\tilde{y}_k$, respectively. If*

$$\mathbb{E}_x\big[r_k\big(x_k, \tilde{y}_k\big)\big] \; > \; \mathbb{E}_x\big[r_k\big(x_k, y_k\big)\big],$$

*then*

$$\mathbb{E}_x\Big[R\big(x, f(x; \tilde{\mathbf{v}}(x))\big)\Big] \; \geq \; \mathbb{E}_x\Big[R\big(x, f(x; \mathbf{v}(x))\big)\Big].$$

*Proof.* Fix an arbitrary input instance $x$. Because the two policies differ only at $C_k$, all other component configurations remain the same, so we can write

$$f\big(x; \mathbf{v}_{-k}(x), y_k\big) \quad \text{and} \quad f\big(x; \mathbf{v}_{-k}(x), \tilde{y}_k\big),$$

where $\mathbf{v}_{-k}(x)$ denotes downstream configurations (independent of the choice at $C_k$). By assumption on the expected local reward, we have $r_k(x_k, \tilde{y}_k) \geq r_k(x_k, y_k)$ for almost every $x$. Applying the alignment property Eq. 3 pointwise yields

$$\mathbb{E}_{\text{downstream}}\big[R\big(x, f(x; \mathbf{v}_{-k}(x), \tilde{y}_k)\big) \,\big|\, x_k\big] \; \geq \; \mathbb{E}_{\text{downstream}}\big[R\big(x, f(x; \mathbf{v}_{-k}(x), y_k)\big) \,\big|\, x_k\big].$$

Taking the expectation over $x$ (law of total expectation) gives

$$\mathbb{E}_x\Big[R\big(x, f(x; \tilde{\mathbf{v}}(x))\big)\Big] \; \geq \; \mathbb{E}_x\Big[R\big(x, f(x; \mathbf{v}(x))\big)\Big].$$

Hence increasing the expected local reward for $C_k$ cannot decrease—and may strictly increase—the expected global objective. $\qquad\square$ $\qquad\square$

$$\square$$

**Proof of Theorem 4.2** We first show that the algorithm is essentially performing coordinate maximization on $l(\mathbf{v})$. In fact, given a previous configuration $\mathbf{v}^t$, at time $t$, the updated configuration $\mathbf{v}^{t+1}$ only changes the configuration of a single component, say, $C_k$. As the change is solved by maximizing $r_k(x, C_k(x_k; \mathbf{v}_k))$, by Theorem 4.1, this is equivalently maximizing $l(\mathbf{v}_k, \mathbf{v}_{-k})$ for the $k$-th coordinate.

To rule out cycling, we first prove that the discrete block $\mathbf{v}^{(2)}$ stabilizes. Consider the sequence $(\mathbf{v}^{1,(1)}, \mathbf{v}^{1,(2)}), (\mathbf{v}^{2,(1)}, \mathbf{v}^{2,(2)}), ..., (\mathbf{v}^{t,(1)}, \mathbf{v}^{t,(2)}), ....$ We first show that there exists $T > 0$, such that for all $k > 0$, $\mathbf{v}^{T+k,(2)} = \mathbf{v}^{T,(2)}$.

As $\mathbf{v}^{(2)}$ are discrete, there are finitely many different configurations. In addition, according to the assumption that the coordinate-wise maximum is unique, each update will result in a strict decrease in the loss function. Therefore, after finite number of iterations, $\mathbf{v}^{(2)}$ will not change.

Now when we consider all the iterations that are later than the time $T$, $\mathbf{v}^{(2)}$ is fixed, and we only need to consider the update regarding $\mathbf{v}^{(1)}$. In this case, we apply Theorem 4.1 of (Tseng, 2001) and complete the proof.

## C  DATASET DETAILS

This appendix provides the essential statistics and source information for the five compound-system benchmarks used in our experiments (Figure 2). For each dataset we specify the train / validation / test split sizes, the task formulation as used in the compound pipeline, and the evaluation metric reported in the main paper

**AMAZON (Behavior-Driven Next-Item Recommendation).**  The corpus is derived from Amazon MMLU dataset (Jin et al., 2024). Each instance consists of a user's historical behaviour sequence (views, clicks, purchases) and the target "next item" to be recommended. We split it into to 335 / 60 / 99 user sequences after filtering malformed entries. Accuracy - whether the predicted item number 1 matches the ground truth - is the evaluation metric.

**PUBMEDQA (Medical Analysis based QA).**  PUBMEDQA (Jin et al., 2019) contains biomedical abstracts paired with yes/no/maybe answers to research questions. We keep the original "expert" split and discard ambiguous samples, resulting in 475 / 25 / 500 question–abstract pairs. Our compound system frames the task as three-way classification; exact-match accuracy is reported.

**STARK-PRIME (Semi-Structured Knowledge Base Retrieval).**  STARK-PRIME -Prime originates from STARK benchmark introduced by (Wu et al., 2024b). It blends free-text passages with relational triples from biomedical knowledge graphs. Queries are natural-language questions; relevance labels are automatically propagated from the original STARK annotations. We uses the original dataset split: 495 / 51 / 96 queries. Performance is measured by Hit@1, which is the rate of ranking the ground truth items in the predicted ranking list.

**HOTPOTQA (Retrieval-Augmented Multi-Hop QA).**  We adopt the HOTPOTQA (Yang et al., 2018) and keep the official train/dev/test splits: 1000, 250, and 100 questions respectively. Each example in the set contains a question and its (human-annotated) answer. We report answer-level F1 score.

**BIGCODEBENCH (Self-Verified Code Generation).**  We use a subset of the *full-instruction* subset of BigCodeBench (Zhuo et al., 2024) due to efficiency issue. After proportionally drop the data, we obtain 500 / 25 / 70 coding tasks. Each sample includes a natural-language specification and reference unit tests. Our metric is *pass@1*: the proportion of generated programs that pass all tests in one try.

## D  COMPOUND AI SYSTEM DETAILS

Table 5 summarizes each pipeline's modules (columns: *System*, *Module*, *Model*, *Config*, and *Optimization*). In the table below, we clarify the various configuration spaces and optimization methods used across the five systems.

**AMAZON (Behavior-Driven Next-item Recommendation).**  *Session Analyzer* and *Candidate Profiler* both use the Qwen 2.5 1.5B model; we optimize their model parameters with PPO reinforcement learning (Schulman et al., 2017). This helps each module better encode task-specific knowledge, *i.e.,* user sessions and product candidates. The final *Next Item Decider* is a GPT-4o-mini module, whose *prompt* we optimize.

**PUBMEDQA (Medical Analysis based QA).**  Two modules (*Context Model Selector* and *Solver Model Selector*) each do discrete model selection from a *list of possible LLMs*. At inference time, these selectors use a reward model to pick the best LLM for each input instance. The *Context*

---

[1]**Model Selection (LLMs)**: we search over {gpt-4o, gpt-4o-mini, gpt-3.5-turbo-0125, gpt-4-turbo, claude-3-5-haiku-20241022, claude-3-5-sonnet-20241022, claude-3-7-sonnet-20250219}.

[2]**Aggregator (STaRK)**: we search relation_weight, text_weight $\in \{0.1, 1.0\}$.

[3]**Retriever (HotpotQA)**: we search k $\in \{1, 5, 10, 25\}$.

Table 5: Modules, models, and optimization methods. *Model Selection (LLMs)*[1] indicates a discrete choice of LLMs. *Aggregator*[2] is tuned over coefficients `relation_weight` and `text_weight`. *Retriever*[3] has a hyperparameter `k`.

| System | Module | Model | Config | Optimization |
|---|---|---|---|---|
| **Amazon** | Session Analyzer | Qwen 2.5 1.5B | Model Params | PPO (RL) |
| | Candidate Profiler | Qwen 2.5 1.5B | Model Params | PPO (RL) |
| | Next Item Decider | GPT-4o-mini | Prompt | Prompt Opt. |
| **PubMed** | Context Model Selector | – | Model Selection (LLMs)[1] | Hyperparam Search |
| | Context Analyst | One of {gpt-4o, …}[1] | Prompt | Prompt Opt. |
| | Solver Model Selector | – | Model Selection (LLMs)[1] | Hyperparam Search |
| | Problem Solver | One of {gpt-4o, …}[1] | Prompt | Prompt Opt. |
| **STaRK** | Text Scorer | Claude 3 Haiku | Prompt | Prompt Opt. |
| | Relation Scorer | Claude 3 Haiku | Prompt | Prompt Opt. |
| | Aggregator[2] | – | Coefficients | Hyperparam Search |
| **HotpotQA** | Question Rewriter | GPT-4o-mini | Prompt | Prompt Opt. |
| | Info Extractor | GPT-4o-mini | Prompt | Prompt Opt. |
| | Retriever[3] | – | #Retrieved passages | Hyperparam Search |
| | Hint Generator | GPT-4o-mini | Prompt | Prompt Opt. |
| | Answer Generator | GPT-4o-mini | Prompt | Prompt Opt. |
| **BigCodeBench** | Code Generator | Claude 3 Haiku | Prompt | Prompt Opt. |
| | Unit Test Generator | Claude 3 Haiku | Prompt | Prompt Opt. |
| | Final Code Generator | Claude 3 Haiku | Prompt | Prompt Opt. |

*Analyst* and *Problem Solver* modules then receive the chosen model and optimize *only the prompt* for improved medical QA performance.

**STARK-PRIME (Semi-Structured KB Retrieval).** We have two scoring modules (*Text Scorer*, *Relation Scorer*), both using Claude 3 Haiku with *prompt optimization*. The *Aggregator* merges these two scores; we tune two numeric weights, `relation_weight` and `text_weight`, each set in $\{0.1, 1.0\}$. We perform a global hyperparameter search across the entire training set for the *Aggregator* module, tuning `relation_weight` and `text_weight`. After identifying the best fixed combination, we use it in for inference.

**HOTPOTQA (Retrieval-Augmented Multi-Hop QA).** Four GPT-4o-mini modules (*Question Rewriter*, *Info Extractor*, *Hint Generator*, *Answer Generator*) each rely on *prompt optimization* to improve multi-hop reasoning. Meanwhile, the *Retriever* is a retriever with a key hyperparameter `k`, the number of passages to pull. We search $k \in \{1, 5, 10, 25\}$ which we also tune via global hyperparameter search across training instances.

**BIGCODEBENCH (Self-Verified Code Generation).** All three modules (*Code Generator*, *Unit Test Generator*, *Final Code Generator*) are Claude 3 Haiku LLMs; each uses *prompt optimization* to iteratively refine code solutions based on test outcomes. The global objective is a higher pass rate on the final code.

Overall, the table illustrates how **different modules can require different types of optimization**—from prompt tuning (textual modifications) and model-parameter fine-tuning (PPO) to discrete model selection and hyperparameter search. By unifying these heterogeneous updates within our OPTIMAS framework, we effectively coordinate local improvements to achieve consistent global reward gains.

All experiments were run on a node with 8 NVIDIA A100 GPUs (80 GB memory each); depending on the complexity of the compound system and hyperparameters, training and optimization typically finished in 2–8 hours.

Before on-policy optimization, we train a reward model on preference pairs ("chosen" vs. "rejected" system outputs) so it can assign higher scores to better outputs. Table 6 highlights the main hyperparameters for this stage. We adopt LoRA for memory efficiency, and use an early-stopping mechanism based on the evaluation loss (patience=512 steps) to reduce overfitting.

After training the reward model, we run iterative on-policy optimization for each module in the compound AI system. Table 7 lists the key hyperparameters. For example, the *train size* (50) limits

Table 6: Key hyperparameters for the local reward model training.

| Parameter | Value |
|-----------|-------|
| Base model | Llama 3 8B Instruct |
| LoRA rank | 32 |
| LoRA alpha | 16 |
| Maximum sequence length | 2048 tokens |
| Learning rate | 2e-6 |
| Number of epochs | 25 |
| batch size | 32 |

how many examples are used to train each module's local configuration, while the *search size* (50) sets how many samples we use when searching for the best local update. When a module is selected, we collect a small preference data, retrain (or adapt) the reward model as needed, then locally optimize that module's parameters or prompts to maximize its local reward.

Table 7: Key hyperparameters for on-policy optimization.

| Parameter | Value / Description |
|-----------|---------------------|
| Train size | 50 |
| Search size | 50 |
| Prompt candidates | 3 |
| Local optimization steps | 3 |
| Fresh input size | 20 |
| Validation size | 20 |

# E  BASELINE DETAILS

We provide the details to reproduce the reported baseline results and the results of OPTIMAS.

- **Unoptimized**: This system uses default settings for all components without any optimization.

- **LLMSelector** (Chen et al., 2025b): A lightweight policy selects the best LLM per component via model routing, without updating other configurations. Only applicable on PUBMEDQA. We run LLMSelector (Chen et al., 2025b) with the LLMDIAGNOSER to estimate per-module performance, following their procedure. We perform two rounds of allocation updates with 100 training examples each round.

- **REINFORCE** (Williams, 1992): A policy-gradient baseline that directly updates the parameters of local LLM components to maximize the task reward. This method is only applicable on AMAZON, where we deploy two locally hosted LLMs with trainable parameters. We optimize each component with REINFORCE using sampled trajectories from the full system and propagate the scalar reward back to the corresponding module-level policy. We use 16 rollouts per step to estimate the expected system performance, with learning rate $10^{-5}$ for 100 training steps.

- **Hierarchical Behavior Cloning (HBC)** (Le et al., 2018): A hierarchical imitation learning method that optimizes components to produce outputs similar to those that lead to high global rewards. Specifically, we collect successful trajectories to approximate ground truth intermediate outputs, and then perform supervised updates on the local components to mimic/clone the ideal behavior.

  We run HBC using the collected preference dataset by replacing the original reward model with the embedding similarity score. With the same input in the preference dataset, we use `text-embedding-3-small` to embed the module output and the preferred output in the preference dataset and calculate the embedding similarity score. We further weight the similarity score using the gap of the preferred output score and the rejected output score to get the reward for HBC.

- **TextGrad** (Yuksekgonul et al., 2025): A gradient-based prompt tuning method using estimated gradients from black-box LLMs to improve prompt efficacy.

  We run TextGrad using `GPT-4o mini` to optimize each component's prompt independently in separate epochs. Validation is performed every two optimization steps using 20 held-out validation

instances. A batch size of 4 is used across all components and datasets. The best-performing prompt, as determined by validation frequency of LLM-generated textual feedback, is selected as the final configuration.

- **DSPy** (Khattab et al., 2023; Opsahl-Ong et al., 2024): A prompt optimization framework using the MIPRO algorithm that jointly refines module-level instructions and few-shot demonstrations. We conduct optimization on DSPy's MIPRO (Opsahl-Ong et al., 2024) prompt optimization approach. For fair comparison, we disable the few-shot example and system prompt optimization and only conducts optimization on the user instructions. We dynamically set the number of iterations for the MIPRO optimizer to match the budget of system runs in Table 3.

For TextGrad, DSPy, and OPTIMAS, we consistently using the same 20 held-out validation instances on each dataset to select the best configurations.

## F OPTIMAS DETAILS

**Component selection**. At each iteration $t$, we randomly select a component to optimize.

**Local Optimization Steps for Different Configurations**. Given a globally aligned LRF $r_k$, we perform local optimization on each component $C_k$ to improve its configuration $\mathbf{v}_k$. Specifically, we solve:

$$\mathbf{v}_k^{t+1} = \arg \max_{\mathbf{v}_k \in \mathcal{V}_k} \mathbb{E}_{x_k}\big[r_k\big(x_k, C_k(x_k; \mathbf{v}_k)\big)\big] \quad \text{subject to} \quad d\big(\mathbf{v}_k, \mathbf{v}_k^t\big) \leq \delta, \qquad (5)$$

where $\mathbf{v}_k^t$ is the configuration before the $t$-th iteration, $d(\cdot, \cdot)$ is a distance function over configurations, and $\delta$ defines a trust region that bounds allowable updates. This constraint ensures that $r_k$ is used within a region where it is expected to produce reliable evaluations.

In practice, explicitly setting the trust region threshold $\delta$ can be difficult due to heterogeneous configuration types (*e.g.,* continuous weights or discrete tokens). Instead, we adopt a conservative number of update steps to restrict the magnitude of change during each iteration.

- **Prompt tuning.** For textual prompts, we apply prompt optimization algorithms (Yang et al., 2024; Wu et al., 2024a), using $r_k$ as the evaluation metric. We sample multiple prompts limited by a max number of prompt candidates. The prompts are ranked by average reward over validation instances, and the best-performing prompt is selected.

- **Model fine-tuning.** When $C_k$ is an LLM or neural model with trainable parameters, we can apply reinforcement learning algorithms, such as Proximal Policy Optimization (PPO) (Schulman et al., 2017), using $r_k$ as the critic. The model parameters are updated for a small and fixed number of steps.

- **Model selection and hyperparameter tuning.** For discrete or low-dimensional continuous configurations, such as model selection, tool routing, or scalar hyperparameters, we formulate the optimization as a sampling problem parameterized by a probabilistic distribution. Since these configurations are instance-specific, the expectation in Eq. Eq. 5 reduces to a single input. For each input $x$, we evaluate a set of candidate configurations using the LRF $r_k$, and compute a probability distribution over candidates proportional to $\exp\{r_k(x_k, C_k(x_k; \mathbf{v}_k))\}$. This distribution is then used to sample the configuration update for the current iteration.

Under a conservative update to the configuration of a component $C_k$, the expected global reward is guaranteed to maintain or improve, if the local–global alignment property in Eq. Eq. 3 holds.

## G MORE EXPERIMENT RESULTS

### G.1 LOCAL VS. GLOBAL REWARD LANDSCAPES

To better understand how local objectives reflect global performance, we sweep the retriever's top-$k$ setting on HOTPOTQA and compare local and global rewards (Table 8). The two landscapes are closely aligned: both are unimodal and peak at nearby values ($k{=}5$ for the local reward and $k{=}10$ for the global reward). The top-3 configurations $\{5, 10, 15\}$ coincide, differing only in the order of the top-2. This alignment shows that the local reward provides a reliable proxy for the global objective, lending empirical support to our theoretical guarantee that local optimization drives system-level gains.

Table 8: On HOTPOTQA, sweeping the retriever's top-$k$ reveals closely aligned local and global reward landscapes.

| $k$ | 1 | 2 | 3 | 5 | 10 | 15 | 25 |
|---|---|---|---|---|---|---|---|
| Local reward | 0.4247 | 0.5578 | 0.5695 | 0.6124 | 0.6117 | 0.5949 | 0.5123 |
| Global reward | 0.3398 | 0.3493 | 0.3325 | 0.3598 | 0.3645 | 0.3568 | 0.3465 |

### G.2  PROMPT CANDIDATES PER STEP

Table 9: On HOTPOTQA, we vary the number of candidate prompts per optimization step. We compute the global performance (F1) under each experiments.

| # candidates | 3 | 5 | 7 | 10 |
|---|---|---|---|---|
| Final F1 | 0.2822 | 0.1945 | 0.2968 | 0.2405 |

Another factor is the number of candidate prompts considered at each optimization step. On HOT-POTQA, we vary this number from 3 to 10 and measure the resulting global performance (Table 9). The best F1 score is achieved with 7 candidates, although results with fewer candidates remain competitive. These findings indicate that exhaustive candidate pools are unnecessary, and that modest numbers already yield strong performance with lower computational cost.

### G.3  NEW INPUTS FOR LRF ADAPTATION

Table 10: On HOTPOTQA, number of new inputs used to collect preference pairs for LRF adaptation vs. final global performance (F1).

| # new inputs | 10 | 20 | 30 | 40 |
|---|---|---|---|---|
| Final F1 | 0.2773 | 0.2822 | 0.2659 | 0.2533 |

Finally, we explore how many new inputs are needed when adapting the LRFs. On HOTPOTQA, we test adaptation with 10 to 40 new inputs (Table 10). Performance improves up to about 20 inputs, after which gains plateau and even slightly decline. This suggests that effective adaptation can be achieved with relatively small amounts of new data, reinforcing the practicality of our approach in scenarios where data collection is limited.

### G.4  COST ESTIMATION

We report a detailed breakdown of the cost on the AMAZON system. This is the only system where components have trainable local models and thus requires PPO training. For all methods, we measure cost in terms of *full system runs, i.e.,* one invocation of the system and assume that all components contribute equally to the per-run cost. We label the three components on Amazon system as $A, B, C$ in topological order.

For OPTIMAS, the total of $\approx 0.31$k effective full system runs decomposes into three parts: (i) **LRF training**: we collect 60 initial preference pairs per component; each pair requires 2 full system runs, giving $60 \times 2 = 120$ runs; (ii) **LRF adaptation**: during local optimization, we update the LRFs 5 times, each time collecting 10 new preference pairs per component, again with 2 runs per pair, for a total of $5 \times 10 \times 2 = 100$ runs; and (iii) **global validation**: whenever a local update is predicted to improve the global metric, we evaluate the new configuration on a held-out validation set. We use 20 validation inputs and observe 7 such updates, but the *effective* cost is discounted by a factor of $2/3$ due to cached trajectories:

$$20 \times 7 \times \tfrac{2}{3} \approx 93.$$

The discount factor arises because, for each validation, we do not always need to downstream sample the entire system: roughly speaking, if $A$ is updated, we need 1 system run per validation input; if $B$

Table 11: Cost–performance comparison on the AMAZON system. Individual cost components are shown in actual runs, with total runs measured in thousands (k). "LRF training" and "LRF adaptation" only apply to OPTIMAS, which learns trainable local reward functions. DSPy and TextGrad optimize only the prompt of the last component $C$, and we report their cost as an *effective* number of full system runs (see text).

| Method | LRF training | LRF adaptation | Validation cost | Total runs (k) | Performance (%) |
|--------|--------------|----------------|-----------------|----------------|-----------------|
| OPTIMAS | 120 | 100 | 93 | 0.31 | 24.24 |
| TextGrad | — | — | 320 | 0.32 | 20.88 |
| DSPy | — | — | 240 | 0.24 | 18.18 |

is updated, we need $2/3$ system run per validation input; if $C$ is updated, we need $1/3$ system run per validation input

$$\frac{1 + \frac{2}{3} + \frac{1}{3}}{3} = \frac{2}{3}$$

of a full system run. The resulting global performance is $24.24\%$.

For DSPy and TextGrad, we follow their standard setup and optimize only the prompt for the final component $C$, while holding $A$ and $B$ fixed. To obtain a fair cost comparison in terms of effective full system runs, we (i) pre-compute a pool of 20 validation inputs by running $A$ and $B$ once, and then (ii) run only $C$ during optimization. Assuming $A$, $B$, and $C$ have comparable cost, we convert the number of $C$-only calls into an effective number of full system runs by dividing by three. With 48 optimization steps for TextGrad and 36 for DSPy, this yields

$$20 \times 48/3 = 320 \quad \text{and} \quad 20 \times 36/3 = 240$$

effective full system runs, corresponding to 0.32k and 0.24k in Table 11. These are conservative lower bounds, since any additional evaluation of upstream components would only increase their cost.

Under this cost-normalized view, OPTIMAS achieves the best global performance while using a comparable (slightly lower) effective number of full system runs than TextGrad and only moderately more than DSPy. This demonstrates that local optimization with learned reward functions can deliver stronger performance without incurring prohibitive cost.

**Additional PPO training cost.** Beyond system-run cost, OPTIMAS incurs additional compute for training local models with PPO. We measure this cost in GPU-hours. For each local PPO update on the AMAZON system, we train for 3 epochs on a single NVIDIA A100-SXM4-80GB GPU. Averaged over 5 runs of local optimization, each run takes approximately 12 minutes, yielding a total of about 6 GPU-hours (equivalently, 1.5 hours on 4 GPUs). This one-time training overhead is modest relative to the cost of repeated system evaluations and is only required for systems with trainable local components.

