# OpenReview forum: "Optimas: Optimizing Compound AI Systems with Globally Aligned Local Rewards"
_ICLR.cc/2026/Conference — ICLR 2026 Poster_

### Official Review · Reviewer_azQy · 2025-10-30

**Soundness:** 2
**Presentation:** 3
**Contribution:** 2
**Rating:** 4
**Confidence:** 4

**Summary:**

The paper presents a method to optimize an agentic system comprising of multiple LLMs and other components. To do so, it first elicits preferences data for each local component and artifically get the preference label that "aligns" with the global system output, in the sense that if response A gets better system reward, then response A is preferred over B locally.

This preference elicitation process is done many times, and each LLM in the system is progressively adjusted/fine-tuned so the system performance improves.

**Strengths:**

1. The presentation of the paper is good. I could quickly understand what their method is about.
2. Their method makes sense from an intuitive point of view - since we cannot do end-to-end learning of the system (due to perhaps, non-differentiable components), we can slowly adjust each component so that it improves the overall system (by looking at how each local step affects the system output. Despite so, there are some flaws in this approach (see weaknesses).
3. Experimental results are comprehensive (I do have some questions regarding some of the set up and validity).

**Weaknesses:**

1. The method is intuitive. However, it has several flaws. The biggest flaw is that it seems to be performing some form of local parameter updates with respect to a non-differentiable reward signal. As far as I know, this has the same flavour as popular gradient-free training such as REINFORCE. Why didn't the authors mention such approaches or use it as the baselines (I'm quite sure we can use it as a baselines).
2. In addition, what makes this method slightly worse than prior approaches such as REINFORCE is that it optimizes each component in a round-robin/random fashion. This means the approach cannot be globally optimal because we are adjusting one component/subset of parameters at a time right? It is misleading to claim that their approach guarantees optimality in this case (we need some extra convex assumption of the optimization landscape).
3. In addition, I feel some of the experiment set up is contrived. For instance, how would a single strong model perform in answering the task directly, instead of creating a complicated system to answer it? This seems like a simple baseline that should be discussed. In addition, it seems like the local rewards have a positive correlation with the global rewards - how does simply improving the local models directly (locally) or using stronger local models fare as a baselines?
4. Lastly, could the authors clarify what they meant that we are reducing the number of system calls? Everytime you try to elicit a preference of the local component, we need to do a forward pass of the system to check the global rewards, is this correct? So we definitely need to accumulate many system calls eventually right?

**Questions:**

See my weakness section above.

---

> ### Author Response · Authors · 2025-11-21
>
> We thank the reviewer for the extensive comments,we clarify our settings and add additional experiments below:
>
> ---
>
> ## **[Comment 1.1:] ”it seems to be performing some form of local parameter updates with respect to a non-differentiable reward signal”**
>
> We firstly want to clarify that
> - The goal of Optimas is not just optimizing model parameters, the learned Local Reward Functions (LRFs) can also be applied to prompt optimization, hyperparameter search, and model selection. For example, in the Amazon system, there’re two local LLMs and one API-based LLM, where the last one requires prompt optimization. In the PubMed system, we optimize two model selectors and one API-based LLMs.
> - Secondly, the reward signal comes from the learned LRF, which is a differentiable neural network that we trained on collected preference data.
>
> ---
>
> ## **[Comment 1.2] ”mention REINFORCE or use it as the baselines”**
> We thank the reviewer for this suggestion. We clarify that REINFORCE can only be applied when the component has differentiable parameters, which limits its applicability in compound AI systems with heterogeneous configurations. For example some other possible components could be:
> - **Textual prompts** (discrete token sequences)
> - **Model selection** (categorical API choices: gpt-4o, Claude, etc.)
> - **Black-box LLM APIs** (no gradient/parameter access)
> - **Hyperparameters** (often discrete: k ∈ {1, 5, 10, 25})
>
> However, as the **Amazon system** does contain trainable local models (two Qwen 2.5 1.5B LLMs), REINFORCE can actually optimize the model parameters. We apologize for not including this baseline initially and provide the results below.
> We use 16 rollouts per step to estimate the expected system performance as the non-differentiable reward signal, with learning rate $10^{-5}$ for 100 training steps.
>
> **Results:**
>
> | Method       | Amazon Product Rec. (Acc.) |
> |-------------|-----------------------------|
> | Unoptimized | 21.21 ± 3.78               |
> | HBC         | 21.55 ± 2.07               |
> | TextGrad    | 20.88 ± 3.53               |
> | DSPy        | 18.18 ± 0.82               |
> | **REINFORCE** |  **21.89 ± 2.65**   |
> | **OPTIMAS** | **24.24 ± 0.82**           |
>
> We do want to mention that since REINFORCE needs collecting the reward signal from  full trajectory via downstream Monte Carlo sampling, it requires more than 3 times the data than Optimas, which uses LRFs for the optimization and avoids downstream sampling during local optimization.
>
> We added REINFORCE to our related work and added the results and discussions in our experiment section. Please see our revision with the highlighted modifications.
>
> ---
>
> ## **[Comment 2] ”guarantees optimality”**
> Thank you for raising this important point.
>
> We clarify that our algorithm does not converge to the global optimum in the general non-convex loss case. In fact, Theorem 4.2 states that our algorithm **converges to a component-wise maximum**, and the subsequent discussion clarifies that if the objective additionally satisfies the Polyak–Łojasiewicz condition (PL) then the algorithm is guaranteed to converge to the global maximum.
>
> Moreover, we note that **this limitation is shared by many standard reinforcement-learning algorithms**: for example, the classic REINFORCE algorithm in a general non-convex Markov Decision Process does not guarantee reaching a global optimum unless additional structural assumptions are imposed. Thus, the fact that our method uses block/round-robin updates does not introduce a novel “weak-point” in isolation. It inherits the standard non-convex limitation found in policy-gradient style methods.
>
> ---
>
> ## **[Comment 3.1] "experiment set up..This seems like a simple baseline that should be discussed.”`**
>
> Thank you for this concern. We did make sure the compound AI system outperformed a single LLM baseline at the first place when we designed these systems.
>
> For example, on HotpotQA, many questions require retrieving relevant information from the internet, and a single LLM that directly outputs answers is hard to obtain a high accuracy. Similarly, multiple LLMs on the BigCodeBench system enable LLMs to reflect based on unit tests and modify the results for higher pass rate. In general, such compound AI systems have been proven to yield better results, which is consistent in our cases. Specifically, we provide a single LLM result and compare with our (unoptimized) compound AI system:
>
> || Amazon (Acc.) | PubMedQA (Acc.) | STaRK-Prime (MRR) | HotpotQA (F1) | BigCodeBench  (Pass Rate) |
> |--|--|--|--|--|--|
> | Single LLM  | $20.20 ± 1.43$| $54.13 ± 2.73$| $0.00 ± 0.00$| $21.58 ± 1.24$| $35.47 ± 0.34$|
> | (Unoptimized) Compound AI system | $21.21 ± 3.78$| $57.46 ± 0.75$| $40.73 ± 0.64$| $33.80 ± 1.51$| $36.67 ± 1.35$|
>
> Notice that STaRK-Prime requires complex multi-stage tool usage, where single LLMs achieve an accuracy of 0%. We hope our clarification and evidence justified the importance of using compound AI systems in complex tasks.

---

> ### Author Response · Authors · 2025-11-21
> **Official Comment by Authors (Continue)**
>
> ## **[Comment 3.2] "how does simply improving the local models directly (locally) or using stronger local models fare as a baselines?"**
>
> - Firstly, Optimas is able to improve local models, for example, through PPO training.
>
> - Secondly, we would like to respectfully highlight our contribution's scope: **Optimas is an optimization framework that improves compound AI systems**, regardless of how they compare to other alternatives. This is analogous to how optimization papers (e.g., Adam for neural networks, Bayesian optimization for hyperparameters) focus on *improving* a given system rather than *justifying* the neural network or system choice. This point is related to the reviewer's comments "a single strong model" and "(why not) using stronger local models as a baselines?". Essentially, Optimas addresses: "Given a compound AI system, how do we optimize it?" rather than "Should we use compound systems vs. single LLMs vs some other alternative?". Therefore, to validate the effective of Optimas, we show that, across five systems (where some systems already have very strong local models), Optimas is able to further improve the system performance significantly compared to the Unoptimized baseline and other optimization methods.
>
> ---
>
> ## **[Comment 4] "clarify what they meant that we are reducing the number of system calls?"**
>
> For Optimas, the system calls occur in two places: 1) optimization, and 2) collecting preference pairs.
> - **Optimization**: We first explain how the system calls are reduced during optimization. For an intuitive example, let’s say our system consists of three sequential components: $A \rightarrow B \rightarrow C$, and we are performing PPO optimization for component $A$. Assume PPO runs for 15 steps with a batch size of 16, resulting in a total of 240 inputs to be evaluated.
>   - **With the local reward functions:** For Optimas, we can obtain the reward directly after each output from $A$ (without executing $B$ and $C$) requiring only $1/3 \times 240 = 80$ equivalent system runs (assuming $A, B$, and $C$ have equal computational cost).
>   - **Without the local reward function functions.** For other methods, we must forward each of the 240 outputs from $A$ through both $B$ and $C$ to evaluate the full system reward. If $B$ and $C$ are stochastic, this typically involves multiple forward passes (denoted by $\ell$) per $A$ output to obtain a reliable reward estimate. The additional cost becomes approximately $2/3 \times 240 \times \ell$ system runs.
> - **Collecting preference pairs**:
>   - While our method does require a preference dataset to train the reward function, **the data requirement is modest**. For `Stage 1: Initial reward modeling` (`Section 4.2`), typically around 50 preference pairs.
>   - And to clarify, **we don’t need to elicit a preference dataset every time** during each optimization step. We only need such adaptation when the configuration of a component is updated (because when the system is unchanged, the LRFs are already aligned from previous preference training). Therefore, referring to Figure 4, if around 5 components are updated throughout the optimization steps, we only need to conduct adaptation around 5 times. Each time, we typically construct 10~20 preference pairs. In other words, we do accumulate system calls but eventually the total number is moderate.
> - Lastly, in Appendix `G.1`, we show that **the amount of preference data does not need to be large to achieve good local-global alignment**. And the cost of this step is usually smaller than the cost of full system runs required without local reward functions.
>
> ---
>
> ## **Summary**
>
> We thank you for the extensive review and insights!
>
> We provided
> - Experimental comparison with REINFORCE,
> - Clarifications on the component-wise optimality,
> - Experiments and clarifications on the experimental setup with compound AI systems, and
> - Clarifications on the system run cost of Optimas.
>
> We are happy to further engage if any of the above points remains unclear. Feel free to let us know.
>
> We will also be glad to know if any of your concerns are addressed, and if so, we would sincerely appreciate your reconsideration of our work. Thank you once again for your valuable insights!

---

> ### Comment · Reviewer_azQy · 2025-11-26
>
> Thank you for the responses.
>
> 1. Firstly, regarding the local optimality of their method. I would say local optimality is certainly a weakness that it suffers from. I know other methods have the same weakness as well, but it wouldn't be fair to say that it is of no issue because what we are doing here is applying PPO to each component in a round-robin manner (or in isolation). PPO of course is only optimal under certain assumptions. This is no problem. But my comment was about applying it in a round-robin manner - that is certainly something that is only introduced in this work, and is a weakness (and not inherited from existing works).
>
> 2. Second, when I say "how does simply improving the local models directly (locally) or using stronger local models fare as a baselines?", I am referring to iteratively improving the local models using simple local optimization approaches and comparing with your approach. Why do you say that this is an invalid baseline? Am I misunderstanding something here? I am not really asking to change the system at all. We can also view this as swapping out a local component for a strong (generally speaking) model (e.g., changing a 7B model to a 14B model). This is certainly considered in the author's system right? Since they mentioned doing model selection.
>
> 3. My last point relates to the point on number of system queries. Let's say you have a system of A -> B -> C. When you generate preference pairs of B, how do you elicit the global preference of this pair? This requires more system passes right?
>
> 4. In addition, how do you actually do a system pass with this preference pair of B? You just pass it to C, correct? This works if we have only a single chain of dependency. What if you have 1) **A -> B** 2) **A -> C**, and **(B, C) -> D** (So D receives inputs from B and C), how do you elicit a system preference over a preference pair generated from B? You are missing the inputs from C if you want to do a system query forward pass. How do we actually elicit the global preference then?

---

> ### Author Response · Authors · 2025-11-27
>
> Thank you for the timely responses! Great questions and here we further address the concerns:
>
> ---
>
> ## **[Further Question 1] ”Local optimization in a round-robin manner”**:
>
> We respectfully clarify that the use of block / round-robin updates in our method does not necessarily constitute a “weak-point” beyond the usual limitations encountered in non-convex optimization. Specifically:
>
>   - As we mentioned and also agreed by the reviewer, the limitation of converging only to local (or component-wise) maxima in general non-convex settings is not introduced by our use of block updates per se — it is a **standard limitation** for nearly all methods that tackle non-convex objectives. Whether one updates all parameters jointly (full gradient / joint update) or in blocks, absent convexity (or PL/KL, or other strong structural assumptions), one cannot guarantee reaching the global optimum.
>   - Therefore, just because we are applying our algorithm “in a block/round-robin manner” does not automatically worsen this fundamental limitation. **Block-coordinate methods remain a mainstream and theoretically studied approach even in non-convex settings**. For instance, recent works show that block-coordinate descent (BCD) for structured non-convex optimization can converge to coordinate-wise stationary points, and under certain additional conditions (e.g., error-bound conditions, regularity, or PL/KL-type conditions) can even yield global or stronger convergence guarantees [1].
>
>   In light of this, we believe it might not be accurate to characterize “round-robin block updates” as an additional, distinct weakness of our method beyond those inherent to non-convex optimization. Rather, our block-update design is simply **one practical algorithmic choice for tractability and scalability; the real limitation remains in the general non-convex setting** — a limitation shared by most existing non-convex / policy-gradient / reinforcement-learning methods.
>
> To improve clarity and avoid misunderstanding, we have revised the manuscript to include an explicit remark in the discussion: that block-coordinate updates do not guarantee global optimality in non-convex problems, but this does not constitute a flaw unique to our method as it reflects a standard limitation of non-convex optimization broadly, and that the global convergence guarantee only holds under additional structural assumptions (such as PL, KL, or similar conditions).
>
> ---
>
> ## **[Further Question 2] ”iteratively improving the local models using simple local optimization approaches and comparing with your approach”**
>
> Our apologies for misunderstanding your previous question and thank you for the clarification! We now better understand your question regarding the baselines.
>
> (1) **"iteratively improving the local models using simple local optimization approaches"**
>
> We agree that if ground-truth labels exist for every component, we should simply optimize them locally using supervised learning. However, the central challenge in Compound AI Systems is that we typically **lack ground-truth labels** for intermediate components. For example, in a Query Rewriter -> Retriever -> Reader system, we have labels for the final Answer, but not for the "perfectly rewritten query."
>
> A potential way to get pseudo ground truth labels is by using successful global trajectories as "ground truth" and performing supervised updates on the local components to mimic that behavior. In our original submission, we have included a **Behavior Cloning-based baseline** which does exactly this. We revised the wording to make this clearer.
>
> Since true local labels are missing, it approximates local optimization by treating successful global trajectories as "ground truth" and performing supervised updates (cloning) on the local components to mimic that behavior. As shown in Table 2, Optimas consistently outperforms it (e.g., +2.69% on Amazon, +10.3% on PubMedQA). This demonstrates that **simply cloning successful traces is insufficient**, likely because it cannot generalize to new inputs or distinguish which part of the trace caused the success, whereas Optimas's learned LRFs provide a more robust signal for optimization.

---

> ### Author Response · Authors · 2025-11-27
>
> (2) **"swapping out a local component for a strong (generally speaking) model"**
>
> We would like to clarify the model selection part in our systems. Model selection is present not as an optimization method, but as **a user-defined configuration** i.e. a design parameter of the system itself, similar to a hyperparameter or a prompt that can be optimized. As discussed in [2], users may explicitly design systems with heterogeneous model candidates because "allocating different LLMs to different modules usually leads to substantially higher performance than allocating the same LLM to all modules". Since model selection is a configuration defined by the user, Optimas treats it as a discrete parameter to be optimized and in our results we show that Optimas outperforms LLMSelector [2], a method specifically for optimizing model choice for different components.
>
>  In this setting, it is not about swapping a "weak" model for a "strong" one; rather, different models possess different strengths. Of course, if a stronger model yields the highest performance for a specific component, Optimas will naturally identify and select it during the optimization process. Conversely, in systems where the models are fixed by the user, Optimas is flexible enough to optimize the remaining configurations, whether via PPO for model parameters or prompt optimization for closed-source models, and outperforms baselines such as behavior cloning, DSPy, TextGrad etc.
>
> ---
>
> ## **[Further Question 3] ”How do you elicit the global preference of the preference pairs of B?”**
>
> Here’s the detailed process:
> - For an initial system input, we collect one full system pass from $A\rightarrow B \rightarrow C$. Let the output of $B$ in this pass be $y_{b1}$.
> - We fixed the input to $B$. Because we want to collect a new output under the same condition.
> - We passed through $B, C$ to get another final output $y_{b2}$.
> - The $(y_{b1}, y_{b2})$ is a preference pair where the winning output is the one that leads to better performance.
>
> The above process is for **a) training the local reward function**. However, it does not necessarily mean it requires more system passes **in total**. This is because, once we train these local reward models, we do not need to do downstream system passes during **b) local optimization**, meaning that we can just get outputs from $B$ and directly obtain the rewards from the local reward model.
>
> In short, the system passes we “saved” in b) local optimization compensate for the extra cost from a) local reward function training. Thus, when we claim to 'reduce system calls,' we refer to the fact that although Optimas does require a small number of system passes to train the reward model, it avoids repeated full-system rollouts during optimization resulting in significantly higher data efficiency, achieving better performance with fewer total system samples compared to baselines.
>
> ---
>
> ## **[Further Question 4] ”How do you elicit a system preference over a preference pair generated from B, where 1) $A \rightarrow B$ 2) $A \rightarrow C$, and $(B, C) \rightarrow D$?”**
>
> Great question. In fact, we won’t miss inputs to $D$ from $C$. Following the last answer, we go through the process again:
>  - For an initial system input, we collect one full system pass from $1) A \rightarrow B$ $2) A \rightarrow C$, and $(B, C) \rightarrow D$. Let the output of $B$ in this pass be $y_{b1}$.
> - We fixed all the intermediate outputs from a node $N$, if there doesn’t exist a path from $N$ to $B$. Since $C$ is one of such nodes, the output from $C$ will be fixed. This is consistent with the previous example where we want to collect a new output under **the same condition**. In our paper, this corresponding to Line 215, where  we state
>
>   ---
>
>   $v_{−k}$ denotes the configurations of all downstream components (those that directly or indirectly receive information originating from $C_k$
>
>   ---
>
>   indicating that for non-downstream components, the outputs from them are fixed. We further made it clear in our updated manuscript.
>
> - We passed through $B$ for a new output $y_{b2}$, along with the same output from $C$ in the initial system pass, to get another final system output from $D$.
> - The $(y_{b1}, y_{b2})$ is a preference pair where the winning output is the one that leads to better performance.
>
> Note that all of the systems in our experiments are beyond the single-chain structures, where we apply the above processes to collect preference data. For even more complex systems, we believe the above steps are also applicable. Hope these are clear!
>
> ---

---

> > ### Author Response · Authors · 2025-11-27
> > **Summary**
> >
> > Thank you again for the time and care you put into this review! We have used your feedback to greatly improve our paper by running new experiments, and updating the text for clarity. We appreciate your earlier suggestion to include REINFORCE as a baseline; incorporating this experiment has strengthened our empirical evaluation. We hope these additional detailed clarifications regarding local optimality, model selection configurations, and system efficiency fully address your remaining concerns.
> >
> >
> > ---
> >
> > [1] Block Coordinate Descent Methods for Structured Nonconvex Optimization with Nonseparable Constraints: Optimality Conditions and Global Convergence. Zhijie Yuan, Ganzhao Yuan, Lei Sun.
> >
> >  [2] Optimizing model selection for compound ai systems Lingjiao Chen, Jared Quincy Davis, Boris Hanin, Peter Bailis, Matei Zaharia, James Zou, and Ion Stoica

---

> > ### Comment · Reviewer_azQy · 2025-11-28
> >
> > Thank you for the responses. The responses mostly answered my questions.
> >
> > >  Question 4 & Question 3 [About how to elicit preferences without affecting other previous components]
> >
> > Oh I think this answers my question. Initially I though adjusting parameters will influence some components in front of the component. But it seems like this isn't the case. But yes, it still requires a forward pass from the component of interest (but not a full system pass).
> >
> > > Question 1 about round-robin.
> >
> > Thanks for answering this question. I guess the round-robin is actually why the optimization approach is feasible. If not, we need to someone do joint-preference elicitation (need to get preferences of multiple components all at once and it no longer become a simple pair-wise comparison, which blows up the optimization dimension). it's kinda weakness, but there's no free lunch (just like gradient descent is a local optimization approach) and we are trading off global system optimality for efficiency.
> >
> > > Question 2 about local optimization.
> >
> > Actually I still think local optimization is still a very possible baseline that real-world practitioners will use. For instance, an owner of a system definitely will try to find ways to improve each local component (with some simple data set etc), and see what kind of performance gains are possible. _Some_ experiment results in the paper (that you pointed out) suggest that local optimization does not outperform the end-to-end optimization. I'd think this point should be emphasized in the paper.
> >
> > **At the moment, my questions are answered. I will increase my score to 6 (however, I think I cannot edit my score at the moment, so I will revisit this later).**
> >
> > To summarize, the strengths of the paper:
> > 1. A novel method to optimize end-to-end agentic systems with automatic preference elicitation. Some approaches exists in previous literature (as cited in the paper), but does something different (e.g., using Bayesian Optimization, etc.). I'd say they all have different tradeoffs (actually, it'd be nice to include what tradeoffs are there for different approaches - like, this method's tradeoff is its greediness towards global optimality, while the weakness of REINFORCE is that it cannot optimize certain kinds of non-differentiable parameters etc.)
> > 2. Experiment results support their method (however I have not looked through the code closely).
> >
> > The weakness:
> > 1. It inherits some of the existing weakness of round-robin algorithms (e.g., does not converge to global optimum unless some conditions are met) and DPO (e.g. preferences do not contain enough information). However, the authors have addressed them in writing during the rebuttals.

---

> > > ### Author Response · Authors · 2025-11-28
> > >
> > > We thank the reviewer for the **verbal promise to raise the score from 4 to 6**.
> > >
> > > We are glad that the questions are mostly addressed with a detailed summarization provided by the reviewer.
> > >
> > > **While editing the score and review is unfortunately disabled, we value your time greatly and will make sure to pass this discussion and summary in our final remarks.**
> > >
> > > Thank you,
> > >
> > > Authors of Paper #1810

---

### Official Review · Reviewer_WMHN · 2025-10-30

**Soundness:** 3
**Presentation:** 3
**Contribution:** 3
**Rating:** 6
**Confidence:** 2

**Summary:**

The paper proposes OPTIMAS, a framework to optimize compound AI systems by learning component-wise Local Reward Functions (LRFs) that are trained to be locally–globally aligned with the task’s global reward.

**Strengths:**

1. The paper clearly motivates why compound AI systems are hard (non-differentiable, heterogeneous knobs) and frames a practical local–global alignment objective.

2. The paper unifies optimization across prompts, hyperparameters, model selection, and model parameters within one iterative loop.

3. The paper offers interpretability hooks—probing LRF preferences (e.g., brevity bias matching F1) to explain why updates help.

**Weaknesses:**

While OPTIMAS reduces the number of full system runs, it introduces non-trivial computational overhead from (i) training and adapting the shared 8B-backbone Local Reward Functions (via LoRA) and (ii) generating preference labels that require downstream sampling to estimate expected global rewards. In settings that enable parameter fine-tuning (e.g., PPO for some modules), total token and wall-clock costs may exceed prompt-only baselines. The paper would be strengthened by a cost breakdown (tokens, time, and dollars) that separates LRF training/adaptation, downstream sampling per preference pair, and any PPO steps, and by a cost-normalized comparison to DSPy/TextGrad.

**Questions:**

1. How often does local improvement (per LRF) fail to translate to global improvement in practice, and what diagnostics or safeguards detect misalignment early?

2. How sensitive is performance to the LRF backbone size and adaptation frequency? Could you show accuracy vs. backbone (1B/3B/8B) and vs. adaptation interval?

3. Are there failure cases where local–global alignment deteriorates over long runs? If so, what mitigation (e.g., trust regions, re-labeling frequency) is most effective?

---

> ### Author Response · Authors · 2025-11-21
>
> Thank you for your constructive and thoughtful suggestions touching on the practicalness of our work! Below we address your concerns:
>
> ---
>
> ## **[Comment 1] “computational overhead from training and adapting the shared 8B-backbone Local Reward Functions”**
>
> We appreciate this concern.
>
> - Our experiments in **Appendix G.3** show that, in fact, the computational cost can be further and significantly reduced with **lightweight alternatives**. For your convenience, here’s a summary of Appendix G.3:
>
>   We conducted experiments with 1B and 3B models in addition to the 8B models to evaluate whether model size impacts alignment quality.
>
>   | LRF Backbone Size | Evaluation Accuracy (%) |
>   | :---: | :---: |
>   | 1B | 72.58% |
>   | 3B | 70.04% |
>   | 8B | 71.20% |
>
>   The results suggest that lightweight models are sufficient for this task, making the training and adaptation computationally efficient.
>
> - Moreover, our **backbone is shared** across different components, meaning we are not required to train $N$ backbones for $N$ components, which may require more data and more compute.
>
> We hope the evidence of lightweight alternatives and shared backbone alleviate your concern on computational overhead.
>
> ---
>
> ## **[Comment 2] “Computational overhead from generating preference labels..require downstream sampling to estimate expected global rewards”**
>
> Thanks for the comment!
>
> - We would like to clarify an important point that **all of our optimization baselines require downstream sampling** to obtain supervision on any intermediate outputs.
>
>   For example, DSPy needs downstream sampling to evaluate if a prompt works better; LLMSelector needs downstream sampling to evaluate certain model combinations. The differences between Optimas and these methods are:
>   - We construct ***preference labels*** to train the local reward functions
>   - We maintain these local reward functions to ***continuously learn from preference data***, instead of giving “one-time” evaluation on configurations like prompts or model combinations. Training and adapting LRFs accumulates experience to distinguish good and bad intermediate outputs, and we show that Optimas achieves better data efficiency.
>
> - Moreover,  since we only update one component at a time, we **only need to sample through the partial system**, instead of the entire system. Say we have a system $A\rightarrow B\rightarrow C\rightarrow D$, when generating preference data for $C$’s output, we only need to forward through $D$. While other methods that change all components will need to sample through the entire system.
>
> We hope the above explanations make it clear that while all optimization methods require supervision from downstream sampling, Optimas reduces the cost by maintaining LRFs that continuously learn from preference data and doing only the necessary amount of sampling.
>
> ---
>
> ## **[Comment 3] “In settings that enable parameter fine-tuning (e.g., PPO for some modules), total token and wall-clock costs may exceed prompt-only baselines. ”`**
>
> In our tasks, we need to optimize systems beyond prompt optimization. For example, PubMed systems have two modules that require optimizing for model selection, and one requires optimize prompt.  In the Amazon system, $A\rightarrow C\leftarrow B(\leftarrow A)$, $A$ and $B$ require parameter fine-tuning and $C$ requires prompt optimization. Here, optimas is able to optimize $A, B, C$, while DSPy or TextGrad can only be applied to $C$.
>
> While it’s likely that the token cost of Optimas will exceed DSPy or TextGrad (because it additionally optimizes $A, B$), the “advantage” of DSPy or TextGrad comes from their limitation that **they cannot be applied** to $A$ and $B$, which can be **a huge barrier** when the bottleneck is at $A$ or/and $B$.
>
> In fact, our work is motivated by this limitation and aims to improve system performance with low additional costs. Even in HotpotQA systems where all the modules are prompt optimizable, we show that Optimas requires the fewest system runs (Table 3).
>
> ---

---

> ### Author Response · Authors · 2025-11-21
>
> ## **[Comment 4] “The paper would be strengthened by a cost breakdown ... and by a cost-normalized comparison to DSPy/TextGrad. ”**
>
> Thank you for this suggestion. We provide a detailed cost breakdown below for the **Amazon system**. Note that not all systems require PPO training, only the Amazon dataset has trainable local models. We give the breakdown below.
>
> - **Data efficiency during LRF training and validation** (assuming equal cost for $A, B, C$):
>
>
>   ---
>
>   | Method     | LRF training (runs)                           | LRF adaptation (runs)                                  | Validation cost (runs)    | Total system runs (k) | Performance|
>   |:----|:----|:----|:----|:----|:----|
>   | **Optimas** | 60 initial pairs/component × 2 runs/pair = 120 | 5 updates × 10 new pairs/component × 2 runs/pair = 100 | 20 val examples × 7 updates × (2/3) discount from caching ≈ 93     | 0.31| 24.24%     |
>   | **TextGrad** | –| –                                                     | 20 val examples × 48 steps ÷ 3 (only optimize C) = 320| 0.32| 20.88%     |
>   | **DSPy**    | –| –| 20 val examples × 36 steps ÷ 3 (only optimize C) = 240| 0.24| 18.18%     |
>
>   ---
>
>
>
>   "Validation cost" is for validating if any configuration change leads to global improvement. For both DSPy and TextGrad, the above estimate is a strict lower bound. Since DSPy/TextGrad only optimize the prompt for the last component $C$, we collect a set of 20 output variants from $A, B$ as the inputs to $C$. We set an early stop step of 20 for both methods, which results in 48 and 36 total steps.
>
>
> - **PPO Optimization time cost**:
> Since we are training local models with the LRFs, we require additional computation. We measure the cost by Hour$\times$GPU. For each PPO optimization, we set the number of epochs to be 3 and we measure the time cost on a single NVIDIA A100-SXM4-80GB. On average across 5 runs of local optimization, each run takes around 12 minutes. Throughout the entire training, the compute cost on the Amazon System is 6 Hour$\times$GPU (equivalently, 1.5 hours on 4 GPUs).
>
> We added the above breakdown in our `Appendix G.6`. Hope this gives a clear view on the cost.
>
>   ---
>
>
> ## **[Question 1.1] ”How often does local improvement (per LRF) fail to translate to global improvement in practice?"**
>
> Great question! In practice, the success rates of local improvements that led to global improvements vary on different datasets and components.
>
> Take the Amazon system for an example again, $71.4$% of local improvements led to global improvements on validation sets. On the HotpotQA system, we observe that only $20$% of local improvements from Keyword Extractor led to global improvement, while the other $80$% led to the same global performance. However, $67.7$% of local improvements from the answer generator led to global improvements.
>
> It would be an interesting future direction to sample components with higher success rates to conduct local optimization more frequently.
>
> ---
>
> ## **[Question 1.2] ”What diagnostics or safeguards detect misalignment early?"**
>
> Similar to standard reward model training, we split the preference data to train and eval splits, and detect misalignment based on the eval performance.
>
> ---
>
> ## **[Question 2] "How sensitive is performance to the LRF backbone size and adaptation frequency?"**
>
> - **LRF backbone size**: Please see the answer in Comment 1.
> - **Adaptation frequency**: When the system configuration is unchanged (e.g., when local optimization doesn’t find a better configuration), there’s no need for adaptation since LRFs are already aligned from previous preference modeling. Therefore, we only need to adapt the LRFs when any one of the components is updated. In other words, we don’t need to control adaptation frequency explicitly. We add clarification to `Section 4.2 Stage 2: Online reward function adaptation`.
>
> ---
>
> ## **[Question 3] ”Are there failure cases where local–global alignment deteriorates over long runs? If so, what mitigation is most effective?**
>
> Thanks for the insightful question! Previously, we observed some overfitting during adaptation due to training on a small batch of new preference data. We mitigate such deterioration using a buffer of previously generated preference data to stabilize training (as described in `Stage 2: Online reward function adaptation` in `Section 4.2`) similar to standard experience replay in off-policy RL.
>
> ---
>
> ## **Summary**
> We hope our responses effectively address your concerns, specifically, we clarify the following
> - Optimas uses lightweight and shared backbones for LRFs to mitigate the computational overhead.
> - All methods require downstream sampling, while Optimas improves data efficiency with learnable LRFs and avoids running the entire system every single time.
> - A breakdown on system runs to understand the cost of Optimas.
>
> Moreover, we answer questions about sensitivity and reliability during training.
>
> Thank you for these great comments again!

---

### Official Review · Reviewer_FLrM · 2025-11-01

**Soundness:** 4
**Presentation:** 4
**Contribution:** 4
**Rating:** 8
**Confidence:** 3

**Summary:**

The paper proposes an effective and efficient framework, **OPTIMAS**, for optimizing compound AI systems with heterogeneous configurations (e.g., prompts, model parameters, and hyperparameters) as a whole. The key idea is to maintain a **Local Reward Function (LRF)** for each component. Before optimizing a component, the corresponding LRF is updated using preference data so that answers preferred by the global reward are also preferred locally (achieving **local–global alignment**). The component is then optimized to maximize its LRF reward.

Theoretically, the authors show that under certain regularity conditions, training an LRF to prefer outputs associated with higher global reward aligns it with the global reward, and optimizing a component with the LRF is equivalent to optimizing it using the global reward. They further show that, under additional assumptions, the optimization converges to a component-wise maximum.

Empirically, the method achieves consistent and substantial improvements across five real-world compound AI systems over five baselines (4 other methods and 1 unoptimized baseline). The authors also validate the mechanism itself: local optimization improves global reward, the trained LRFs exhibit stronger alignment with global rewards than a simple LLM judge, and higher alignments lead to higher global rewards. Additionally, they analyze interpretability and efficiency of the method.

**Strengths:**

1. The presentation is clear and well-organized.
2. The addressed problem—optimizing a compound AI system that could have heterogeneous configurations as a whole rather than its parts independently—is timely and practically important.
3. Theoretical analysis under some assumptions provides partial guarantees for the effectiveness of the method.
4. Empirical evaluation is comprehensive and convincingly supports the method’s effectiveness and mechanism.

**Weaknesses:**

There appear to be some problems in Theorem 4.1 and Lemma B.1. Specifically, Equation 4 defines a loss with a leading minus sign, so Theorem 4.1 should refer to the **minimizer** of Equation. 4, not the **maximizer**. Similarly, in Lemma B.1 the optimization should be formulated as an **argmax** of the expected log-likelihood. The proof shall remain conceptually sound after correcting these signs.

Minor typo issues:
- Line 94: “optimized” → “optimize”
- Line 119: add comma before “While”
- Line 249: “Eq. equation 3”
- Line 256: “r_j” → “r_i”

Otherwise, the paper is technically solid.

**Questions:**

In Theorem 4.1, should it be the 'minimizer' of equation 4  instead of the 'maximizer'. And in Lemma B.1 should it be formed as an argmax problem instead of argmin?

---

> ### Author Response · Authors · 2025-11-21
>
> We sincerely appreciate the reviewer’s approval!
>
> Thanks for the careful reading and spotting the typos in the theoretical proofs! We have corrected the following:
>
> ---
>
> | | Before | After |
> |:---|:-----|:-----|
> |In Theorem 4.1 | “the **maximizer** of equation 4” | “the **minimizer** of equation 4” |
> |In Lemma B.1 | $\text{arg} \textbf{min} _{p} E[log(σ1(y · p(x)))]$  | $\text{arg} \textbf{max} _{p} E[log(σ1(y · p(x)))]$ |
>
> We also fixed the other typos you mentioned in the uploaded revision. Thank you again!

---

### Author Response · Authors · 2025-12-03
**A Summary: Positive Evaluations from Reviewers**

We deeply appreciate all of the reviewers for their efforts in giving us detailed and constructive reviews.

**As of now, all reviewers are expressing a positive evaluation on our work**, specifically:

- ### **`Reviewer #1 FLrM`** (**Score: 8**)
- ### **`Reviewer #2 WMHN`** (**Score: 6**)
- ### **`Reviewer #3 azQy`** (**Score: 4** $\rightarrow$ **6**)
  ```
  At the moment, my questions are answered. I will increase my score to 6.
  ```

---

## **Initial review**
We particularly appreciate the acknowledgment of

- **Clear and well-organized presentation** (`Reviewer #1, #2, #3`)
- **Strong motivation**  ("timely and practically important problem") (`Reviewer #1, #2`)
- **Intuitive** (`Reviewer #3`) and **unified approach** (`Reviewer #2`)
- **Comprehensive evaluation and convinicng empirical effectiveness** (`Reviewer #1, #3`)
- **Theoretical support** (`Reviewer #1`) and **Interpretability** (`Reviewer #2`)  of our method

---

## **Rebuttal**

Here's a summary table about the main concerns we addressed and the updates on the manuscript:

|Concerns|Action or Updates on paper|
|:---|:---|
|Cost breakdown (`Reviewer #2`)|We provide a detailed comparison and added it to `Appendix G.6`|
|Computational overhead (`Reviewer #2`) |We clarify and highlight the exisiting experiments in `Appendix G.3`|
|Comparison with REINFORCE (`Reviewer #3`) | We conduct experiments for REINFORCE and added it to our `Related work and Table 2`|
|Clarification on component-wise optimality and eliciting the preference pairs (`Reviewer #3`) |We clarify that the global optimality is not guaranteed due to the nature of the non-convex optimization problem in `Section 4.4` and make the descriptions about constructing preference clear in `Section 4.2`|

---

## **Finally**

We have highlighted all updates in green for easy reference. We believe Optimas unifies the optimization on compound AI systems with heterogeneous configurations and presents valuable progress.


We sincerely thank everyone again for their time, efforts, and insights in helping make our work better!

---

### Meta-Review · Area_Chair_V8Ls · 2026-01-08

**Summary:**

The initial reviews were generally positive regarding the motivation and unified approach, but also raised concerns about the computational overhead/cost and the absence of key baselines (i.e., REINFORCE). During the rebuttal and discussion period, the author provided a detailed cost breakdown and efficiency analysis (to Reviewer WMHN) and added the requested REINFORCE and Single-LLM baselines (to Reviewer azQy). Since the major concerns were addressed (Reviewer asQy had changed the score from 4 to 6, being positive), AC is to recommend acceptance at this stage.

**Reviewer Concerns:**

Addressed:
1. Missing baselines: The authors added experiments comparing Optimas against REINFORCE (for trainable components) and a Single LLM baseline. Results showed Optimas outperforms REINFORCE (e.g., +2.35% on Amazon) and significantly outperforms single LLMs.
2. Computational overhead: The authors provided a detailed breakdown of system runs, LRF training, and adaptation costs. They demonstrated that Optimas is comparable to or more efficient than DSPy/TextGrad.
3. Technical details: the authors clarified the mechanism for eliciting preferences in complex graphs. Additionally, they addressed the concern regarding global optimality, acknowledging that while their block-coordinate update method converges to component-wise maxima in non-convex settings, this is a standard limitation shared by policy-gradient methods rather than a unique flaw of their approach.

Still Outstanding: No outstanding concerns.

**Reviewer Scores:**

Reviewer FLrM:  remains 8.
Reviewer WMHN: remains 6.
Reviewer azQy: has changed from 4 to 6.

---

### Decision · Program_Chairs · 2026-01-26

Accept (Poster)